# Benefits of Permutation-Equivariance in Auction Mechanisms

**Tian Qin**[1, 2*]    **Fengxiang He**[2*]    **Dingfeng Shi**[2]    **Wenbing Huang**[3, 4]    **Dacheng Tao**[2]

[1]University of Science and Technology of China    [2]JD Explore Academy, JD.com Inc.
[3]Gaoling School of Artificial Intelligence, Renmin University of China
[4]Beijing Key Laboratory of Big Data Management and Analysis Methods
`tqin@mail.ustc.edu.cn, fengxiang.f.he@gmail.com,`
`shidingfeng@buaa.edu.cn, hwenbing@126.com, dacheng.tao@gmail.com`

## Abstract

Designing an incentive-compatible auction mechanism that maximizes the auctioneer's revenue while minimizes the bidders' ex-post regret is an important yet intricate problem in economics. Remarkable progress has been achieved through learning the optimal auction mechanism by neural networks. In this paper, we consider the popular *additive valuation* and *symmetric valuation* setting; *i.e.*, the valuation for a set of items is defined as the sum of all items' valuations in the set, and the valuation distribution is invariant when the bidders and/or the items are permutated. We prove that permutation-equivariant neural networks have significant advantages: the permutation-equivariance decreases the expected ex-post regret, improves the model generalizability, while maintains the expected revenue invariant. This implies that the permutation-equivariance helps approach the theoretically optimal *dominant strategy incentive compatible* condition, and reduces the required sample complexity for desired generalization. Extensive experiments fully support our theory. To our best knowledge, this is the first work towards understanding the benefits of permutation-equivariance in auction mechanisms.

## 1 Introduction

Optimal auction design [30] has wide applications in economics, including computational advertising [18], resource allocation [16], and supply chain [4]. In an auction, every bidder has a private valuation profile over all items, and accordingly, submits her bid profile. An auctioneer collects the bids from all bidders, and determines a feasible item allocation to the bidders as well as the prices the bidders need to pay. Consequently, every bidder receives her utility. From the auctioneer's perspective, the optimal auction mechanism is required to maximize her *revenue*, defined as the sum of all bidders' payments. From the aspect of the bidders, the optimal auction mechanism needs to incentivize every bidder to bid their truthful valuation profiles (truthful bidding). This is summarized as the *dominant strategy incentive compatible* (DSIC) condition; *i.e.*, truthful bidding is always the dominant strategy for every bidder [25].

The optimal auction mechanism can be approximated via neural networks [12, 29, 10]. The "approximation error", or the "distance" to the DSIC condition, is usually measured by the *ex-post regret*, defined as the gap between the bidder's utility of truthful bidding and the utility when her bid profile is only the best to herself (selfish bidding), while the bid profiles of all other bidders are fixed in both cases [12]. When a bidder's ex-post regret is 0, truthful bidding is her dominant strategy. Therefore, the optimal auction design can be modeled as a linear programming problem, where the object is to maximize the expected revenue subject to the expected ex-post regret being 0 for all bidders [12].

---

*Both authors contributed equally.

36th Conference on Neural Information Processing Systems (NeurIPS 2022).

Another major consideration in learning the optimal mechanism is the generalizability to unseen data, usually measured by the *generalization bound*, *i.e.*, the upper bound of the gap between the expected revenue/ex-post regret and their empirical counterparts on the training data [12].

In this paper, we consider the popular setting of *additive valuation* and *symmetric valuation* [12, 29, 10]. The additive valuation condition defines the valuation for a set of items as the sum of the valuations for all items in this set. The symmetric valuation condition assumes the joint distribution of all bidders' valuation profiles to be invariant when bidders and/or items are permutated. This setting covers many applications in practice. For example, when items are independent, the additive valuation condition holds. Moreover, if the auction is anonymous or the order of the items is not prior-known, the symmetric valuation condition holds.

We demonstrate that permutation-equivariant models have significant advantages in learning the optimal auction mechanism as follows. **(1)** We prove that the permutation-equivariance in auction mechanisms decreases the expected ex-post regret while maintaining the expected revenue invariant. Conversely, and equivalently, the permutation-equivariance promises a larger expected revenue, when the expected ex-post regret is fixed. **(2)** We show that the permutation-equivariance of auction mechanisms reduces the required sample complexity for desirable generalizability. We prove that the $l_{\infty,1}$-distance between any two mechanisms in the mechanism space decreases when they are projected to the permutation-equivariant mechanism (sub-)space. This smaller distance implies a smaller covering number of the permutation-equivariant mechanism space, which further leads to a small generalization bound [12].

We further provide an explanation for the learning process of non-permutation-equivariant neural networks (NPE-NNs). In learning the optimal auction mechanism by an NPE-NN, we show that an extra positive term exists in the quadratic penalty of the ex-post regret based on the result **(1)**. This term serves as a regularizer to penalize the "non-permutation-equivariance". Moreover, this regularizer also interferes the revenue maximization, and thus affects the learning performance of NPE-NNs. This further explains the advantages of permutation-equivariance in auction design.

Experiments in extensive auction settings are conducted to verify our theory. We design permutation-equivariant versions of RegretNet (RegretNet-PE and RegretNet-test) by projecting the RegretNet [12] to the permutation-equivariant mechanism space in the training and test stage respectively. The empirical results show that permutation-equivariance helps: (1) significantly improve the revenue while maintain the same ex-post regret; (2) record the same revenue with a significantly lower ex-post regret; and (3) narrow the generalization gaps between the training ex-post regret and its test counterpart. These results fully support our theory.

**Related works.** Myerson completely solves the optimal auction design problem in one-item auctions [23]. However, solutions are not clear when the number of bidders/items exceeds one [12]. Initial attempts have been presented on the characterization of optimal auction mechanisms [20, 26, 9] and algorithmic solutions [1, 2, 27]. Remarkable advances have been made in the required sample complexity for learning the optimal auction mechanism in various settings, including single-item auctions [5, 21, 16], multi-item single-bidder auctions [11], combinatorial auctions [3, 32], and allocation mechanisms [24]. Machine learning-based auction design (automated auction design) have obtained considerable progress [6, 7, 31]. The optimal auction design is modeled as a linear programming problem [6, 7]. However, early works suffer from the scalability issues that the number of the constraints grows exponentially when the bidder number and the item numbers increase. To address this issue, recent works propose to learn the optimal auction mechanism by deep learning. RegretNet is designed for multi-bidder and multi-item settings [12]. Then, RegretNet is developed to meet more restrictive constraints, such as the budget condition [14] and the certifying strategyproof condition [8]. Rahme *et al.* [29] propose the first equivariant neural network-based auction mechanism design method with significant empirical advantages. Ivanov *et al.* [17] propose a RegretFormer which (1) introduces attention layers to RegretNet to learn permutation-equivariant auction mechanisms, and (2) adopts a new interpretable loss function to control the revenue-regret trade-off. Duan *et al.* [10] extend the applicable domain to contextual settings. All these works make remarkable contributions in designing new algorithms from the empirical aspect only. However, the theoretical foundations are still elusive. To our best knowledge, our paper is the first work on theoretically studying the benefits of permutation equivariance in auction design via deep learning.

## 2 Notations and Preliminaries

**Auction.** Suppose $n$ bidders are bidding $m$ items in an auction. Every bidder $i$ has her bidder-context (feature) $x_i \in \mathcal{X}$, while every item $j$ is associated with its item-context (feature) $y_j \in \mathcal{Y}$. The bidder $i$ has a private valuation $v_{ij} \in \mathcal{V} \subset \mathbb{R}_{\geq 0}$ for the item $j$, which is sampled from a conditioned distribution $\mathbb{P}(\cdot | x_i, y_j)$. The value profile $v_i = (v_{i1}, \ldots, v_{im})$ is unknown to the auctioneer. For the simplicity, we define $x = (x_1^T, \ldots, x_n^T)^T$, $y = (y_1, \ldots, y_m)$, $v = (v_1^T, \ldots, v_n^T)^T$, $v_{-i} = (v_1^T, \ldots, v_{i-1}^T, v_{i+1}^T, \ldots, v_n^T)^T$, and $(v_i', v_{-i}) = (v_1^T, \ldots, (v_i')^T \ldots, v_n^T)^T$.

Every bidder submits a bid profile $b_i$ to the auctioneer according to her valuation profile. Then, the auctioneer determines a feasible item allocation $g(b, x, y)$ and corresponding payments $p(b, x, y)$ as per an auction mechanism $(g, p)$. Consequently, every bidder receives her utility as

$$u_i(v_i, b, x, y) = \sum_{j=1}^{m} g_{ij}(b, x, y) \cdot v_{ij} - p_i(b, x, y).$$

The auction mechanism $(g, p)$ consists of an allocation rule $g : \mathbb{R}^{n \times m} \times \mathcal{X}^n \times \mathcal{Y}^m \to \mathbb{R}^{n \times m}$ and a payment rule $p : \mathbb{R}^{n \times m} \times \mathcal{X}^n \times \mathcal{Y}^m \to \mathbb{R}^{n \times m}$, where $g_{ij}$ is the probability of allocating item $j$ to the bidder $i$, and $p_i = \sum_{j=1}^{m} p_{ij}$ is the price that the bidder $i$ should pay. To avoid allocating an item over once, the allocation rule is constrained such that $\sum_{i=1}^{n} g_{ij}(b, x, y) \leq 1$ for all $j \in [m]$. Every $v_i$ in our notations can be replaced by $b_i$. Thus, we can define the similar notations: $b_{-i} = (b_1^T, \ldots, b_{i-1}^T, b_{i+1}^T, \ldots, b_n^T)^T$, $(b_i, v_{-i}) = (v_1^T, \ldots, (b_i)^T \ldots, v_n^T)^T$ and $(v_i, b_{-i}) = (b_1^T, \ldots, (v_i)^T \ldots, b_n^T)^T$.

**Optimal auction mechanism.** An auction mechanism $(g, p)$ is defined to be *dominant strategy incentive compatible* (DSIC), if truthful bidding is always a dominant strategy of every bidder; *i.e.*,

$$u_i(v_i, (v_i, b_{-i}), x, y) \geq u_i(v_i, b, x, y),$$

for all $i \in [n]$, $v, b \in \mathcal{V}^{n \times m}$, $x \in \mathcal{X}^n$ and $y \in \mathcal{Y}^m$. In addition, an auction mechanism $(g, p)$ is called *individually rational* (IR), if for any bidder-contexts $x \in \mathcal{X}^n$, any item-contexts $y \in \mathcal{Y}^m$, any bidder $i \in [n]$ $x \in \mathcal{X}^n$, valuation profile and bid profile $v, b \in \mathcal{V}^{n \times m}$, truthful bidding always leads to a non-negative utility, *i.e.*,

$$u_i(v_i, (v_i, b_{-i}), x, y) \geq 0.$$

If an auction mechanism is DSIC and IR, a rational bidder with an obvious dominant strategy will play it (bidding truthfully). Moreover, an optimal auction mechanism is required to maximize the auctioneer's expected revenue $rev = \mathbb{E}_{(v,x,y)} \big[ \sum_{i=1}^{n} p_i(v, x, y) \big]$.

**Auction design.** The ex-post regret $reg_i(v, x, y)$ for the bidder $i$ is defined as

$$\max_{b_i' \in \mathcal{V}^m} u_i(v_i, (b_i', v_{-i}), x, y) - u_i(v_i, v, x, y).$$

An auction mechanism $(g, p)$ is DSIC, if and only if $\sum_{i=1}^{n} reg_i(v, x, y) = 0$ for any value profile $v \in \mathcal{V}^{n \times m}$, bidder-context $x \in \mathcal{X}^n$, and item-context $y \in \mathcal{Y}^m$. Suppose the payment rule $p$ satisfies $p_i(b, x, y) \leq \sum_{j=1}^{m} g_{ij}(b, x, y) b_{ij}$, which implies that each bidder has a non-negative utility. Then, the auction design can be modeled as a linear programming problem that maximizes the expected revenue $\mathbb{E}_{(v,x,y)}[\sum_{i=1}^{n} p_i(v, x, y)]$ subject to the expected ex-post regret $\mathbb{E}_{(v,x,y)}[\sum_{i=1}^{n} reg_i(v, x, y)] = 0$. Without loss of generality, the ex-post regret may refer to the average of all bidders' ex-post regrets.

**Definition 2.1.** Suppose the network's parameter is $\omega$, and the bidder $i$'s empirical payment and ex-post regret are defined as $\frac{1}{L} \sum_{l=1}^{L} p_i^{\omega}(v^{(l)}, x^{(l)}, y^{(l)})$, and $\widehat{reg}_i(\omega) = \frac{1}{L} \sum_{l=1}^{L} reg_i^{\omega}(v^{(l)}, x^{(l)}, y^{(l)})$, where the sample set $\{(v^{(l)}, x^{(l)}, y^{(l)})\}_{l=1}^{L}$ is *i.i.d.* sampled from the following prior distribution, $\mathbb{P}(v, x, y) = \prod_{i,j=1}^{n,m} \mathbb{P}(v_{ij} | x_i, y_j) \mathbb{P}_{X_i}(x_i) \mathbb{P}_{Y_j}(y_j)$.

**Equivariant mapping.** We define a mapping $f$ as *G-equivariant* if $\psi_g \circ f = f \circ \rho_g$ for two chosen group linear representations $\rho$ and $\psi$ and any $g$ in group $G$.

**Definition 2.2** (Permutation-Equivariant Mapping). A permutation-equivariant mapping is defined to be $f : \mathbb{R}^{n \times m} \to \mathbb{R}^{n \times m}$ that for any instance $x \in \mathbb{R}^{n \times m}$, and permutation matrices $\sigma_n \in \mathbb{R}^{n \times n}$ and $\sigma_m \in \mathbb{R}^{m \times m}$, we have $f(\sigma_n x \sigma_m) = \sigma_n f(x) \sigma_m$.

In this paper, we consider the bidder-permutation $\sigma_n \in \mathbb{R}^{n \times n}$ and item-permutation $\sigma_m \in \mathbb{R}^{m \times m}$. Specifically, we define a mapping $f$ is bidder-symmetric or item-symmetric, if $f(\sigma_n x) = \sigma_n f(x)$ or $f(x\sigma_m) = f(x)\sigma_m$, respectively. Moreover, we define an auction mechanism $(g, p)$ as bidder-symmetric or item-symmetric, if the allocation rule $g$ and the payment rule $p$ are both bidder-symmetric or item-symmetric.

**Orbit averaging.** For any feature mapping $f : \mathcal{F} \to \mathcal{G}$, the orbit averaging $\mathcal{Q}$ on $f$ is defined as $\mathcal{Q}f = \frac{1}{|G|} \sum_{g \in G} \psi_g^{-1} \circ f \circ \rho_g$, where $\rho$ and $\psi$ are two chosen group representations acting on the feature spaces $\mathcal{F}$ and $\mathcal{G}$, respectively. Orbit averaging can project any mapping to be equivariant:

**Proposition 2.3.** *Orbit averaging $\mathcal{Q}$ is a projection to the equivariant mapping space $\{f : \psi \circ f = f \circ \rho\}$, i.e., $\psi \circ \mathcal{Q}f = \mathcal{Q}f \circ \rho$ and $\mathcal{Q}^2 = \mathcal{Q}$. In particular, if $f$ is already equivariant, then $\mathcal{Q}f = f$.*

Moreover, $\mathcal{Q}u$ and $\mathcal{Q}reg$ refer to the utility and the ex-post regret induced by $\mathcal{Q}g$ and $\mathcal{Q}p$. For the simplicity, we denote the orbit averagings that modify the auction mechanism to be bidder-symmetric, item-symmetric, and bidder/item-symmetric by bidder averaging $\mathcal{Q}_1$, item averaging $\mathcal{Q}_2$, and bidder-item aggregated averaging $\mathcal{Q}_3$. Besides, a detailed proof of the feasibility of the projected mechanisms can be found in Appendix A.1.

**Hypothesis complexity.** The generalizatbility to unseen data is usually measured by the generalization bound, which depends on the hypothesis set's complexity. To characterize the complexity of the hypothesis set, we introduce the following definitions of *covering number $\mathcal{N}_{\infty,1}$* and its corresponding distance $l_{\infty,1}$. Based on the covering number, we can obtain a generalization bound in Theorem 3.6.

**Definition 2.4** ($l_{\infty,1}$-distance). Let $\mathcal{X}$ be a feature space and $\mathcal{F}$ a space of functions from $\mathcal{X}$ to $\mathbb{R}^n$. The $l_{\infty,1}$-distance on the space $\mathcal{F}$ is defined as $l_{\infty,1}(f, g) = \max_{x \in \mathcal{X}} (\sum_{i=1}^n |f_i(x) - g_i(x)|)$.

**Definition 2.5** (Covering number). Covering number $\mathcal{N}_{\infty,1}(\mathcal{F}, r)$ is the minimum number of balls with radius $r$ that can cover $\mathcal{F}$ under $l_{\infty,1}$-distance.

# 3 Theoretical Results

This section presents the theoretical results. For simplicity, we view $p = (p_1, \ldots, p_n)^T$ as a $n \times 1$ matrix to present the prices the bidders should pay. We first prove that the permutation-equivariance induces the same expected revenue and a smaller expected ex-post regret in Section 3.1. Next in Section 3.2, we prove that the permutation-equivariant mechanism space has a smaller covering number, which promises a smaller required sample complexity and a better generalization. Detailed proofs are omitted from the main text and given in supplementary materials due to space limitation.

## 3.1 Benefits for Revenue and Ex-Post Regret

In this section, we discuss the benefits for the revenue and the ex-post regret in the conditions of bidder-symmetry and item-symmetry separately, and then discuss the benefits when both of them hold. Based on these results, we also study the learning process of non-permutation-equivariant neural networks for auction design.

### 3.1.1 Benefits in the Bidder/Item-Symmetry Condition

When the bidders come from the same distribution, the joint valuation distribution $f$ is invariant under bidder-permutation, *i.e.* $f(\sigma_n v, \sigma_n x, y) = f(v, x, y)$ for any $\sigma_n \in S_n$. Meanwhile, when the items are indistinguishable, the joint distribution $f$ is invariant under item-permutation, *i.e.*, $f(v\sigma_m, x, y\sigma_m) = f(v, x, y)$ for any $\sigma_m \in S_m$. Both conditions do not always hold simultaneously. In this section, we study them separately.

To measure the "non-permutation-equivariance" of the mechanism, we introduce the conception of *regret gap* between the projected mechanism and the original mechanism as below,

$$\Delta.(g, p; v, x, y) = \max_{v' \in \mathcal{V}^{n \times m}} \sum_{i=1}^n u_i(v_i, (v_i', v_{-i}), x, y) - \max_{v' \in \mathcal{V}^{n \times m}} \sum_{i=1}^n [\mathcal{Q}.u]_i(v_i, (v_i', v_{-i}), x, y),$$

where $v$ is the valuation profiles, $v_i$ is the valuation profile of bidder $i$, $x$ is the bidder-context, $y$ is the item-context, and the orbit averaging $\mathcal{Q}.$ can be the bidder averaging $\mathcal{Q}_1$ or the item averaging $\mathcal{Q}_2$.

The bidder averaging $\mathcal{Q}_1$ and the item averaging $\mathcal{Q}_2$ acting on the allocation rule $g$ and the payment rule $p$, respectively, are as below,

$$\mathcal{Q}_1 g(v, x, y) = \frac{1}{n!} \sum_{\sigma_n \in S_n} \sigma_n^{-1} g(\sigma_n v, \sigma_n x, y), \quad \mathcal{Q}_1 p(v, x, y) = \frac{1}{n!} \sum_{\sigma_n \in S_n} \sigma_n^{-1} p(\sigma_n v, \sigma_n x, y),$$

$$\mathcal{Q}_2 g(v, x, y) = \frac{1}{m!} \sum_{\sigma_m \in S_m} g(v\sigma_m, x, y\sigma_m)\sigma_m^{-1}, \quad \text{and} \quad \mathcal{Q}_2 p(v, x, y) = \frac{1}{m!} \sum_{\sigma_m \in S_m} p(v\sigma_m, x, y\sigma_m).$$

We thus can prove the following theorem that characterizes the benefits of permutation-equivariance for revenue and ex-post regret in the condition of bidder/item-symmetry.

**Theorem 3.1** (Benefits for revenue and ex-post regret in the condition of bidder/item-symmetry).
*When the valuation distribution is invariant under permutations of bidders/items, the projected mechanism has the same expected revenue and a smaller expected ex-post regret, that is,*

$$\mathbb{E}_{(v,x,y)}\left[ \sum_{i=1}^n [\mathcal{Q}.p]_i(v,x,y) \right] = \mathbb{E}_{(v,x,y)}\left[ \sum_{i=1}^n p_i(v,x,y) \right], \quad and \tag{1}$$

$$\mathbb{E}_{(v,x,y)}\left[ \sum_{i=1}^n reg_i(v,x,y) \right] - \mathbb{E}_{(v,x,y)}\left[ \sum_{i=1}^n [\mathcal{Q}.reg]_i(v,x,y) \right] = \mathbb{E}_{(v,x,y)}\left[ \Delta.(g,p;v,x,y) \right] \geq 0, \tag{2}$$

*where $p$ is the payment rule, $reg$ is the ex-post regret, and $\mathcal{Q}.$ is the bidder/item averaging.*

A smaller expected ex-post regret implies this mechanism is closer to the *dominant strategy incentive compatible* condition. Conversely, and equivalently, when the expected ex-post regrets are fixed, the projected auction mechanism has a larger expected revenue. For any auction mechanism, in the bidder/item-symmetry condition, we can project it through the bidder/item averaging.

**Remark 3.2.** The mechanism space can be decomposed into the direct sum of the permutation-equivariant mechanism space $\{\mathcal{M} : \mathcal{Q}\mathcal{M} = \mathcal{M}\}$ and the complementary space $\{\mathcal{N} : \mathcal{Q}\mathcal{N} = 0\}$ [13]. Thus, a mechanism $\mathcal{M}$ has a unique decomposition: $\mathcal{M} = \mathcal{Q}\mathcal{M} + \mathcal{N}$. The pure permutation-equivariant part $\mathcal{Q}\mathcal{M}$ contains all and only the "permutation-equivariance" of the mechanism $\mathcal{M}$. The pure non-permutation-equivariant part $\mathcal{N}$ is independent from the permutation-equivairance. In this way, we may study the influence of permutation-equivariance by comparing the mechanism $\mathcal{M}$ and its permutation equivariant part $\mathcal{Q}\mathcal{M}$.

### 3.1.2 Interplay between Bidder-Symmetry and Item-Symmetry.

If the valuation distribution is invariant under both bidder-permutation and item-permutation, we can project the mechanism to be permutation-equivariant with respect to both bidder and item in two steps (by mapping $\mathcal{Q}_1 \circ \mathcal{Q}_2$ or mapping $\mathcal{Q}_2 \circ \mathcal{Q}_1$). Consequently, the projected mechanism has the same expected revenue and a smaller expected ex-post regret. Equivalently, we can also project an auction mechanism to be bidder-symmetric and item-symmetric immediately by the bidder-item aggregated averaging $\mathcal{Q}_3$ as below,

$$\mathcal{Q}_3 g(v, x, y) = \frac{1}{n!m!} \sum_{\sigma_n \in S_n} \sum_{\sigma_m \in S_m} \sigma_n^{-1} g(\sigma_n v \sigma_m, \sigma_n x, y\sigma_m)\sigma_m^{-1}, \quad \text{and}$$

$$\mathcal{Q}_3 p(v, x, y) = \frac{1}{n!m!} \sum_{\sigma_n \in S_n} \sum_{\sigma_m \in S_m} \sigma_n^{-1} p(\sigma_n v \sigma_m, \sigma_n x, y\sigma_m).$$

We can prove that the bidder-item aggregated averaging $\mathcal{Q}_3$ is the composition of the orbit averaging operators $\mathcal{Q}_1$ and $\mathcal{Q}_2$, as shown in the following lemma. This lemma shows that the order of $\mathcal{Q}_1$ and $\mathcal{Q}_2$ would not influence their composition.

**Lemma 3.3.** *The bidder-item aggregated averaging is the composition of bidder averaging and item averaging: $\mathcal{Q}_3 = \mathcal{Q}_1 \circ \mathcal{Q}_2 = \mathcal{Q}_2 \circ \mathcal{Q}_1$.*

Based on this lemma, we can prove the following theorem on the benefits of permutation-equivariance for revenue and ex-post regret in the condition of both bidder-symmetry and item-symmetry.

**Theorem 3.4** (Benefits for revenue and ex-post regret in the condition of both bidder-symmetry and item-symmetry). *When the valuation distribution is invariant under both item-permutation and item-permutation, then the projected mechanism has a same expected revenue and a smaller expected ex-post regret, that is,*

$$\mathbb{E}_{(v,x,y)}\left[\sum_{i=1}^{n}\mathcal{Q}_3 p_i(v,x,y)\right] = \mathbb{E}_{(v,x,y)}\left[\sum_{i=1}^{n} p_i(v,x,y)\right] \text{ and}$$

$$\mathbb{E}_{(v,x,y)}\left[\sum_{i=1}^{n} reg_i(v,x,y)\right] - \mathbb{E}_{(v,x,y)}\left[\sum_{i=1}^{n}[\mathcal{Q}_3 reg]_i(v,x,y)\right] = \mathbb{E}_{(v,x,y)}\left[\Delta_3(g,p;v,x,y)\right] \geq 0,$$

*where $p$ is the payment rule, $reg$ is the ex-post regret, and $\mathcal{Q}_3$ is the bidder-item aggregated averaging.*

The difference between bidder-symmetry and item-symmetry is significant in practice. For example, for a symmetric valuation distribution, when the mechanism is already bidder-symmetric but not item-symmetric, we can project it to be item-symmetric to gain an extra benefit from item-symmetry. That means, the two regret gaps induced by $\mathcal{Q}_1$ and $\mathcal{Q}_2$ are "additive" as below,

$$\Delta_3(g,p;v,x,y) = \Delta_1(g,p;v,x,y) + \Delta_2(\mathcal{Q}_1 g, \mathcal{Q}_1 p;v,x,y).$$

In general, $\mathbb{E}[\Delta_2(g,p;v,x,y)] \neq \mathbb{E}[\Delta_2(\mathcal{Q}_1 g,\mathcal{Q}_1 p;v,x,y)]$ and thus $\mathbb{E}[\Delta_3(g,p;v,x,y)] \neq \mathbb{E}[\Delta_1(g,p;v,x,y)] + \mathbb{E}[\Delta_2(g,p;v,x,y)]$. Thus, the benefits from bidder-symmetry and item-symmetry are "additive" but not strictly "independent".

### 3.1.3 Insights on Training Non-Permutation-Equivariant Mechanism

Because the expected revenue is always the same for the original mechanism and the projected permutation-equivariant mechanism, we only consider the gradient caused by the expected ex-post regret. We can decompose the original expected ex-post regret into the sum of the expected ex-post regret of the projected mechanism and the expectation of the regret gap as below,

$$\mathbb{E}_{(v,x,y)}\left[\sum_{i=1}^{n} reg_i(v,x,y)\right] = \mathbb{E}_{(v,x,y)}\left[\sum_{i=1}^{n}[\mathcal{Q}_3 reg]_i(v,x,y)\right] + \mathbb{E}_{(v,x,y)}\left[\Delta_3(g,p;v,x,y)\right].$$

The regret gap $\Delta_3(g,p;\cdot)$ follows from the "non-permutation-equivariance" of the mechanism $\mathcal{M}$. When the distance $l(\mathcal{M},\mathcal{QM})$ tends to 0, the regret gap converges to 0. When the auction mechanism has a negligible ex-post regret, the expectation of the regret gap is also close to 0. That means, the mechanism is close to being permutation-equivariant. However, even using a symmetric dataset or adopting data augmentation in training, the learned mechanism will not be permutation-equivariant in general [19]. As a result, to achieve negligible ex-post regret, the non-permutation-equivariant models need to learn more samples to approach permutation-equivariance. That is because the non-permutation-equivariant part (expected regret gap) would mislead the gradient of the expected regret but have no benefit to the expected revenue and the expected ex-post regret.

On the other hand, the regret gap can be viewed as a regularizer in the ex-post regret to penalize the "non-permutation-equivariance" of the mechanism. When the optimizer tries to minimize the ex-post regret, the auction mechanism approaches to be permutation-equivariant. Therefore, if the mechanism achieves a negligible ex-post regret, it is almost to be permutation-equivariant. This result can explain why RegretNet struggles to find permutation-equivariant auction mechanisms [29]. However, in complex settings, it will be harder for non-permutation-equivariant models to approach the negligible ex-post regret. It can explain why the permutation-equivariant models show a significant improvement in complex settings, compared with that they have similar performances in simple settings [10, 17], which shows the great importance of adopting permutation-equivariant models in complex settings.

## 3.2 Benefits for Generalization

In this section, we study permutation-equivariance from the aspect of generalizability [22, 15], which characterizes the performance gap of a learned mechanism on collected training data and unseen data.

We first study the covering number of the permutation-equivariant mechanism space. Let $\mathcal{U} = \{u^\omega : \omega \in \Omega\}$ and $\mathcal{P} = \{p^\omega : \omega \in \Omega\}$ be the spaces of all possible utilities and payment rules, and

$\mathcal{Q}.\mathcal{U} = \{\mathcal{Q}.u : u \in \mathcal{U}\}$ and $\mathcal{Q}.\mathcal{P} = \{\mathcal{Q}.p : p \in \mathcal{P}\}$ the spaces of all projected utilities and payment rules. In addition, let $\mathcal{N}_{\infty,1}(\mathcal{U}, r)$ and $\mathcal{N}_{\infty,1}(\mathcal{P}, r)$ be the minimum numbers of balls with radius $r$ that can cover $\mathcal{U}$ and $\mathcal{P}$ under $l_{\infty,1}$-distance, respectively. We obtain the following result, which indicates the projected permutation-equivariant mechanism space has smaller covering numbers.

**Theorem 3.5** (Covering number of the permutation-equivariant mechanism space)**.** *The space of all projected bidder-symmetric mechanisms has smaller covering numbers, that is,*

$$\mathcal{N}_{\infty,1}(\mathcal{Q}_1\mathcal{U}, r) \leq \mathcal{N}_{\infty,1}(\mathcal{U}, r) \quad \text{and} \quad \mathcal{N}_{\infty,1}(\mathcal{Q}_1\mathcal{P}, r) \leq \mathcal{N}_{\infty,1}(\mathcal{P}, r).$$

*The space of all projected item-symmetric mechanisms has smaller covering numbers, that is,*

$$\mathcal{N}_{\infty,1}(\mathcal{Q}_2\mathcal{U}, r) \leq \mathcal{N}_{\infty,1}(\mathcal{U}, r) \quad \text{and} \quad \mathcal{N}_{\infty,1}(\mathcal{Q}_2\mathcal{P}, r) \leq \mathcal{N}_{\infty,1}(\mathcal{P}, r).$$

Intuitively, the orbit averaging $\mathcal{Q}$ narrows the distance between two mechanisms: $l(\mathcal{QM}, \mathcal{QM}') \leq l(\mathcal{M}, \mathcal{M}')$, for any two mechanisms. Then, any $r$-cover $\mathcal{A}$ for space $\mathcal{U}$ or space $\mathcal{P}$ induces an $r$-cover $\mathcal{QA}$ for space $\mathcal{QU}$ or space $\mathcal{QP}$.

Combining with Lemma 3.3, we have the following results,

$$\mathcal{N}_{\infty,1}(\mathcal{Q}_3\mathcal{U}, r) = \mathcal{N}_{\infty,1}(\mathcal{Q}_1\mathcal{Q}_2\mathcal{U}, r) \leq \mathcal{N}_{\infty,1}(\mathcal{Q}_2\mathcal{U}, r) \leq \mathcal{N}_{\infty,1}(\mathcal{U}, r) \text{ and}$$
$$\mathcal{N}_{\infty,1}(\mathcal{Q}_3\mathcal{P}, r) = \mathcal{N}_{\infty,1}(\mathcal{Q}_1\mathcal{Q}_2\mathcal{P}, r) \leq \mathcal{N}_{\infty,1}(\mathcal{Q}_2\mathcal{P}, r) \leq \mathcal{N}_{\infty,1}(\mathcal{P}, r).$$

We then prove that two generalization bounds of permutation-equivariant mechanisms, which characterize the gap between the expected revenue/ex-post regret and their empirical counterparts. Similar generalization results are existing in previous works [10, 12].

**Theorem 3.6** (Generalization bounds of permutation-equivariant mechanisms)**.** *If for any bidder, her valuation satisfies that $v_i(S) \leq 1$ for any $S \subset [m]$, then with probability at least $1 - \delta$, we have the following inequalities with $\epsilon \geq \sqrt{\frac{9n^2}{2L}(\log\frac{4}{\delta} + \max\{\log\mathcal{N}_{\infty,1}(\mathcal{P}, \frac{\epsilon}{3}), log\mathcal{N}_{\infty,1}(\mathcal{U}, \frac{\epsilon}{6})\})}$,*

$$\left| \mathbb{E}\left[ \sum_{i=1}^{n} p_i^{\omega}(v, x, y) \right] - \frac{1}{L} \sum_{l=1}^{L} \sum_{i=1}^{n} p_i^{\omega}(v^{(l)}, x^{(l)}, y^{(l)}) \right| \leq \epsilon, \quad \text{and} \tag{3}$$

$$\left| \mathbb{E}\left[ \sum_{i=1}^{n} reg_i^{\omega}(v, x, y) \right] - \sum_{i=1}^{n} \widehat{reg}_i(\omega) \right| \leq \epsilon, \tag{4}$$

*where $L$ is the number of samples, $\mathcal{U}$ and $\mathcal{P}$ are the spaces of all possible utilities and payment rules.*

Equivalently, we can rewrite this result in the form of the sample complexity,

**Corollary 3.7.** *For any $\epsilon > 0$, $\delta \in (0, 1)$, and mechanism parameter $\omega$, when the sample complexity $L \geq \frac{9n^2}{2\epsilon^2}\left(\log\frac{4}{\delta} + \max\left\{\log\mathcal{N}_{\infty,1}(\mathcal{P}, \frac{\epsilon}{3}), log\mathcal{N}_{\infty,1}(\mathcal{U}, \frac{\epsilon}{6})\right\}\right)$, with probability at least $1 - \delta$, the generalization bounds, eqs. (3) and (4), hold.*

**Remark 3.8.** Combining Theorem 3.5, we have proved that the permutation-equivariance can improve the generalizability.

## 4 Experiments

This section presents our experimental results. More details and results are presented in the supplementary materials.

**Model architecture.** We project RegretNet [12] to the permutation-equivariant mechanism space via employing bidder-item aggregated averaging for the bidder-symmetry and item-symmetry condition. The projected model is called RegretNet-PE. We also project the well-trained RegretNet, called RegretNet-Test. Specifically, RegretNet is an auction mechanism defined as $(g^{\omega}, p^{\omega})$, in which both the allocation rule $g^{\omega}$ and the payment rule $p^{\omega}$ are neural networks that consist of three fully-connected layers, and $\omega$ is the overall model parameter of the auction mechanism. The detailed architecture is given in the supplementary materials.

**Comparison with EquivariantNet.** RegretNet uses two feed-forward fully-connected networks to learn the allocation rule and payment rule, respectively. We denote the weight matrix in the

Table 1: Experimental results. "$n \times m$ Uniform" refers that there are $n$ bidders and $m$ items, and the valuations are i.i.d. drawn from the uniform distribution $U[0, 1]$. To simplify, we multiply all results by a factor of $10^5$ for the ex-post regret and generalization error (GE).

| Method | $2 \times 1$ Uniform | | | $3 \times 1$ Uniform | | | $5 \times 1$ Uniform | | |
|---|---|---|---|---|---|---|---|---|---|
| | Revenue | Regret | GE | Revenue | Regret | GE | Revenue | Regret | GE |
| Optimal | 0.417 | 0 | - | 0.531 | 0 | - | 0.672 | 0 | - |
| RegretNet | 0.415 | 17.4 | 6.00 | 0.535 | 18.3 | 11.4 | 0.658 | 15.9 | 6.40 |
| RegretNet-Test | 0.415 | 16.3 | - | 0.535 | 13.3 | - | 0.658 | 6.50 | - |
| RegretNet-PE | 0.420 | 14.6 | 3.90 | 0.541 | 16.4 | 10.2 | 0.677 | 13.2 | 5.10 |

Table 2: Experimental results. "$n \times m$ Uniform" refers that there are $n$ bidders and $m$ items, and the valuations are i.i.d. sampled from the uniform distribution $U[0, 1]$.

| Method | $1 \times 2$ Uniform | | $2 \times 2$ Uniform | |
|---|---|---|---|---|
| | Revenue | Regret | Revenue | Regret |
| RegretNet | 0.562 | 0.00061 | 0.870 | 0.00070 |
| EquivariantNet | 0.551 | 0.00013 | 0.873 | 0.00100 |
| RegretNet-Test | 0.562 | 0.00052 | 0.870 | 0.00054 |
| RegretNet-PE | 0.563 | 0.00037 | 0.913 | 0.00067 |

layer $\ell$ as $W^{(\ell)}$. Both EquivariantNet and RegretNet-PE inherit the architecture of RegretNet (with some modifications), but utilize different approaches to realize the permutation-equivariance. EquivariantNet applies parameter-sharing in every layer during training, to constrain $W^{(\ell)}$ to be equivariant. In contrast, RegretNet-PE employs orbit averaging to be permutation-equivariant. Specifically, RegretNet-PE adopts a weight matrix $I_K \otimes W^{(\ell)}(\rho_{g_1}^T \ldots \rho_{g_K}^T)^T$ in the first layer, weight matrices $I_K \otimes W^{(\ell)}$ in the following layers, and multiples a matrix $(\rho_{g_1}^{-1}, \ldots, \rho_{g_K}^{-1})$ to the output layer, where $K$ is the scale of the group $G = \{g_1, \ldots, g_K\}$, $\rho_{g_k}$ represents the permutation operator on bidders and items, $I_K$ is an identity matrix, and $\otimes$ is the Kronecker product. It is worth noting that RegretNet-PE is only designed for verifying our theory.

**Auction settings.** We first adopt the two-bidder single-item, two-bidder single-item, three-bidder single-item, and five-bidder single-item settings in the experiments that compare the learned mechanisms with theoretical optimal mechanisms. The optimal auction mechanism for any single-item auction is known [23]. We thus compare the mechanisms leaned by our method with the optimal auction mechanisms in the single-item settings. Also, we compare RegretNet-PE and EquivariantNet in the one-bidder, two-item setting, and the two-bidder, two-item setting. Besides, we employ a multivariate uniform distribution $U[0, 1]^m$ to model the bidder valuation profiles. In all settings, we sample 640,000 data points for training and 5,000 points for test. Due to the space limitation, we place the results of two-bidder five-item and five-bidder three-item settings in Appendix B.2.

**Model training.** We optimize the auction mechanism model via solving the following optimization problem, following the standard settings [12, 29, 10, 17],

$$\mathcal{L}_\rho(\omega, \lambda) = -\frac{1}{L} \sum_{l=1}^{L} \sum_{i=1}^{n} p_i^\omega(v^{(l)}, x^{(l)}, y^{(l)}) + \sum_{i=1}^{n} \lambda_i \widehat{reg}_i(\omega) + \frac{\rho}{2} \Big( \sum_{i=1}^{n} \widehat{reg}_i(\omega) \Big)^2,$$

where $\lambda \in \mathbb{R}^n$ is the Lagrange multiplier and $\rho > 0$ is the factor of the quadratic regularization term. During the training process, the objective function $\mathcal{L}_\rho(\omega, \lambda)$ is minimized via Adam with a learning rate of $0.001$ with respect to the model parameter $\omega$ and the Lagrange multiplier $\lambda$ is updated once in every 100 iterations, until the ex-post regret is smaller than $0.001$. The regularization factor $\rho$ is set to $1.0$ initially and gradually increased along the training process. In calculating the best bid profile $v_i'$ of every bidder $i$, we first randomly initialize the bid profiles once in training and 1,000 times in test, optimize each of them individually via Adam with the same settings, and take the best one as the approximated best bidding.

**Evaluation.** We leverage three metrics to evaluate the performance of the auction mechanism, which are: (1) the empirical revenue $\widehat{rev}$, (2) the empirical ex-post regret averaging across all bidders, *i.e.*, $\widehat{reg} = \frac{1}{n}\sum_{i=1}^{n}\widehat{reg}_i$, and (3) the generalization error defined on top of regrets, *i.e.*, $GE = |\widehat{reg}_{test} - \widehat{reg}_{train}|$, where $\widehat{reg}_{test}$ and $\widehat{reg}_{train}$ are the empirical ex-post regrets during test and training, respectively.

**Computing resource.** The experiments are conducted on 1 GPU (NVIDIA® Tesla® V100 16GB) and 10 CPU cores (Intel® Xeon® Processor E5-2650 v4 @ 2.20GHz).

**Experimental results.** We train a RegretNet and a RegretNet-PE on the training data. The well-trained RegretNet is then projected to be permutation-equivariant, denoted as "RegretNet-Test". The results are collected in Tables 1 and 2.

From Tables 1 and 2, we observe that (1) compared to RegretNet, RegretNet-PE has a significantly higher revenue with a lower ex-post regret, and narrows the generalization gap between the training ex-post regret and its test counterpart; (2) compared to RegretNet, RegretNet-Test receives the same revenue with a significantly lower ex-post regret; and (3) under comparable ex-post regrets, RegretNet-PE has considerably higher revenue than EquivariantNet, while all permutation-equivariant models (RegretNet-Test and RegretNet-PE) can outperform RegretNet. These results show significant benefits of permutation-equivariance on revenue, ex-post regret, and generalizability, which fully supports our theoretical findings in Theorems 3.1, 3.4, and 3.5.

## 5   Conclusion and future works

In this paper, under *additive valuation* and *symmetric valuation* setting, we study the benefits of permutation-equivariance in auction mechanisms in two aspects: a better performance and a better generalization. First, we prove a smaller expected ex-post regret and the same expected revenue when projecting a mechanism to be permutation-equivariant. Next, we propose the permutation-equivariant mechanism space has a smaller covering number, which promises the permutation-equivariant models a better generalization. Extensive experiments are conducted to verify our theoretical results. Our results help understand the optimal auction mechanisms' characterization and the learning processes difference between non-equivariant models and equivariant models.

Beyond the additive valuation setting, an interesting direction is to extend our results to other conditions, including the combinatorial and the unit-demand auctions. Meanwhile, the understanding of the difference in the aspect of the training process between non-equivariant models and equivariant models is still elusive.

**Social impact.** Our results can help understand and design optimal auction mechanisms for symmetric valuation distribution. As a result, our work could inspire more near-optimal auction mechanisms and promote economic growth. No potential negative social impact is identified.

## Acknowledgments and Disclosure of Funding

This work was supported in part by the Major Science and Technology Innovation 2030 "New Generation Artificial Intelligence" Key Project (No. 2021ZD0111700), the National Natural Science Foundation of China (No. 62006137), and the Beijing Outstanding Young Scientist Program (No. BJJWZYJH012019100020098).

We sincerely appreciate Hang Yu, Xiaowen Wei, Kaifan Yang, Shaopeng Fu, and Qingsong Zhang for the valuable comments and the anonymous NeurIPS reviewers for the helpful feedback.

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
