# A  Proofs

This appendix collects all proofs omitted from the main text due to space limitation.

## A.1  Proofs about Orbit Averaging

In this section, we prove Proposition 2.3 and the feasibility of the projected mechanisms. For simplicity, we use the notations $\sigma_n(i)$ and $\sigma_m(j)$ to present the ranks of the bidder $i$ and the item $j$ after bidder-permutation $\sigma_n$ and item-permutation $\sigma_m$, respectively.

**Proof of Proposition 2.3.**   By the definition of orbit averaging $\mathcal{Q}$, we have

$$\mathcal{Q}f \circ \rho_g(x) = \frac{1}{|G|} \sum_{h \in G} \psi_h^{-1} f(\rho_h \rho_g x) = \psi_g \circ \frac{1}{|G|} \sum_{h \in G} \psi_{hg}^{-1} f(\rho_{hg} x) = \psi_g \circ \mathcal{Q}f(x).$$

In addition, if $f$ is equivariant, then we have

$$\mathcal{Q}f = \frac{1}{|G|} \sum_{g \in G} \psi_g^{-1} \circ f \circ \rho_g = \frac{1}{|G|} \sum_{g \in G} \psi_g^{-1} \circ \psi_g \circ f = \frac{1}{|G|} \sum_{g \in G} f = f.$$

Thus, orbit averaging is a projection to equivariant function space and fixes all equivariant functions. In addition, orbit averaging fixes all equivariant functions. That means, every equivariant function can be obtained by orbit averaging. In this sense, every equivariant models are contained in the orbit averaging framework.

**Proof of the feasibility of projected mechanisms.**   We verify all feasibility conditions for the projected mechanisms as follows.

Firstly, for the allocation rule, we have

$$\sum_{i=1}^{n} (\mathcal{Q}_1 g)_{ij}(v, x, y) = \frac{1}{n!} \sum_{i=1}^{n} \sum_{\sigma_n \in S_n} g_{\sigma_n^{-1}(i)j}(\sigma_n v, \sigma_n x, y)$$

$$= \frac{1}{n!} \sum_{\sigma_n \in S_n} \left[ \sum_{i=1}^{n} g_{\sigma_n^{-1}(i)j}(\sigma_n v, \sigma_n x, y) \right] \leq \frac{1}{n!} \sum_{\sigma_n \in S_n} 1 = 1,$$

$$\sum_{i=1}^{n} (\mathcal{Q}_2 g)_{ij}(v, x, y) = \frac{1}{m!} \sum_{i=1}^{n} \sum_{\sigma_m \in S_m} g_{i\sigma_m^{-1}(j)}(v\sigma_m, x, y\sigma_m)$$

$$= \frac{1}{m!} \sum_{\sigma_m \in S_m} \left[ \sum_{i=1}^{n} g_{i\sigma_m^{-1}(j)}(v\sigma_m, x, y\sigma_m) \right] \leq \frac{1}{m!} \sum_{\sigma_m \in S_m} 1 = 1,$$

and

$$\sum_{i=1}^{n} (\mathcal{Q}_3 g)_{ij}(v, x, y) = \frac{1}{n!m!} \sum_{i=1}^{n} \sum_{\sigma_n \in S_n} \sum_{\sigma_m \in S_m} g_{\sigma_n^{-1}(i)\sigma_m^{-1}(j)}(\sigma_n v \sigma_m, \sigma_n x, y\sigma_m)$$

$$= \frac{1}{n!m!} \sum_{\sigma_n \in S_n} \sum_{\sigma_m \in S_m} \left[ \sum_{i=1}^{n} g_{\sigma_n^{-1}(i)\sigma_m^{-1}(j)}(\sigma_n v \sigma_m, \sigma_n x, y\sigma_m) \right] \leq \frac{1}{n!m!} \sum_{\sigma_n \in S_n} \sum_{\sigma_m \in S_m} 1 = 1,$$

thus we know the projected allocation rule is also feasible, which will never allocate one item more than once.

In addition, for the payment rule, we have

$$(\mathcal{Q}_1 p)_i(v, x, y) = \frac{1}{n!} \sum_{\sigma_n \in S_n} p_{\sigma_n^{-1}(i)}(\sigma_n v, \sigma_n x, y)$$

$$\leq \frac{1}{n!} \sum_{\sigma_n \in S_n} \left[ \sum_{j=1}^{m} g_{\sigma_n^{-1}(i)j}(\sigma_n v, \sigma_n x, y)(\sigma_n v)_{\sigma_n^{-1}(i)j} \right]$$

$$= \sum_{j=1}^{m} \left[ \frac{1}{n!} \sum_{\sigma_n \in S_n} g_{\sigma_n^{-1}(i)j}(\sigma_n v, \sigma_n x, y)v_{ij} \right] \leq \sum_{j=1}^{m} (\mathcal{Q}_1 g)_{ij}(v, x, y)v_{ij},$$

$$(\mathcal{Q}_2 p)_i(v, x, y) = \frac{1}{m!} \sum_{\sigma_m \in S_m} p_i(v\sigma_m, x, y\sigma_m)$$

$$\leq \frac{1}{m!} \sum_{\sigma_m \in S_m} \left[ \sum_{j=1}^{m} g_{ij}(v\sigma_m, x, y\sigma_m)(v\sigma_m)_{ij} \right]$$

$$= \sum_{j=1}^{m} \left[ \frac{1}{m!} \sum_{\sigma_m \in S_m} g_{i\sigma_m^{-1}(j)}(v\sigma_m, x, y\sigma_m) v_{ij} \right] \leq \sum_{j=1}^{m} (\mathcal{Q}_2 g)_{ij}(v, x, y) v_{ij},$$

and

$$(\mathcal{Q}_3 p)_i(v, x, y) = \frac{1}{n!m!} \sum_{\sigma_n \in S_n} \sum_{\sigma_m \in S_m} p_{\sigma_n^{-1}(i)}(\sigma_n v \sigma_m, \sigma_n x, y \sigma_m)$$

$$\leq \frac{1}{n!m!} \sum_{\sigma_n \in S_n} \sum_{\sigma_m \in S_m} \left[ \sum_{j=1}^{m} g_{\sigma_n^{-1}(i)j}(\sigma_n v \sigma_m, \sigma_n x, y \sigma_m)(\sigma_n v \sigma_m)_{\sigma_n^{-1}(i)j} \right]$$

$$= \sum_{j=1}^{m} \left[ \frac{1}{n!m!} \sum_{\sigma_n \in S_n} \sum_{\sigma_m \in S_m} g_{\sigma_n^{-1}(i)\sigma_m^{-1}(j)}(\sigma_n v \sigma_m, \sigma_n x, y \sigma_m) v_{ij} \right] \leq \sum_{j=1}^{m} (\mathcal{Q}_3 g)_{ij}(v, x, y) v_{ij},$$

thus, we have completed this proof.

### A.2 Proof of Theorem 3.1

In this section, we proves Theorem 3.1.

*Proof of Theorem 3.1.* We first study the part in the condition of bidder-symmetry; *i.e.*, the orbit averaging $\mathcal{Q}.$ is the bidder averaging $\mathcal{Q}_1$, acting on the allocation rule $g$ and the payment rule $p$ as below,

$$\mathcal{Q}_1 g(v, x, y) = \frac{1}{n!} \sum_{\sigma_n \in S_n} \sigma_n^{-1} g(\sigma_n v, \sigma_n x, y),$$

and

$$\mathcal{Q}_1 p(v, x, y) = \frac{1}{n!} \sum_{\sigma_n \in S_n} \sigma_n^{-1} p(\sigma_n v, \sigma_n x, y).$$

**Step 1:** We first prove that the auction mechanism has the same expected revenue after projection, *i.e.*,

$$\mathop{\mathbb{E}}_{(v,x,y)} \left[ \sum_{i=1}^{n} [\mathcal{Q}_1 p]_i(v, x, y) \right] = \mathop{\mathbb{E}}_{(v,x,y)} \left[ \sum_{i=1}^{n} p_i(v, x, y) \right].$$

Given that for all permutation $\pi$,

$$\sum_{i=1}^{n} p_i = \sum_{i=1}^{n} p_{\pi(i)},$$

we have the following equation,

$$\mathop{\mathbb{E}}_{(v,x,y)} \left[ \sum_{i=1}^{n} \mathcal{Q}_1 p_i(v, x, y) \right] = \mathop{\mathbb{E}}_{(v,x,y)} \left[ \frac{1}{n!} \sum_{i=1}^{n} \sum_{\sigma_n \in S_n} p_{\sigma_n^{-1}(i)}(\sigma_n v, \sigma_n x, y) \right]$$

$$= \mathop{\mathbb{E}}_{(v,x,y)} \left[ \frac{1}{n!} \sum_{i=1}^{n} \sum_{\sigma_n \in S_n} p_i(\sigma_n v, \sigma_n x, y) \right] = \frac{1}{n!} \sum_{i=1}^{n} \sum_{\sigma_n \in S_n} \mathop{\mathbb{E}}_{(v,x,y)} \left[ p_i(\sigma_n v, \sigma_n x, y) \right]$$

$$= \frac{1}{n!} \sum_{i=1}^{n} \sum_{\sigma_n \in S_n} \mathop{\mathbb{E}}_{(v,x,y)} \left[ p_i(v, x, y) \right] = \sum_{i=1}^{n} \mathop{\mathbb{E}}_{(v,x,y)} \left[ p_i(v, x, y) \right].$$

Thus, we complete the first step.

**Step 2:** We then prove that the sum of all bidders' utilities remains the same after projection, *i.e.*,

$$\mathbb{E}_{(v,x,y)}\left[\sum_{i=1}^{n}[\mathcal{Q}_1 u]_i(v_i, v, x, y)\right] = \mathbb{E}_{(v,x,y)}\left[\sum_{i=1}^{n} u_i(v_i, v, x, y)\right].$$

Given that

$$\sum_{i=1}^{n} g_{\pi(i)j} v_{\pi(i)j} = \sum_{i=1}^{n} g_{ij} v_{ij},$$

we have the following equation,

$$\mathbb{E}_{(v,x,y)}\left[\sum_{i=1}^{n}[\mathcal{Q}_1 u]_i(v_i, v, x, y)\right]$$

$$= \mathbb{E}_{(v,x,y)}\left[\sum_{i=1}^{n}\left[\sum_{j=1}^{m}[\mathcal{Q}_1 g]_{ij}(v, x, y) v_{ij} - [\mathcal{Q}_1 p]_i(v, x, y)\right]\right]$$

$$= \mathbb{E}_{(v,x,y)}\left[\sum_{i=1}^{n}\sum_{j=1}^{m}[\mathcal{Q}_1 g]_{ij}(v, x, y) v_{ij}\right] - \mathbb{E}_{(v,x,y)}\left[\sum_{i=1}^{n} p_i(v, x, y)\right]$$

$$= \mathbb{E}_{(v,x,y)}\left[\frac{1}{n!}\sum_{i=1}^{n}\sum_{j=1}^{m}\sum_{\sigma_n \in S_n} g_{\sigma_n^{-1}(i)j}(\sigma_n v, \sigma_n x, y) v_{ij}\right] - \mathbb{E}_{(v,x,y)}\left[\sum_{i=1}^{n} p_i(v, x, y)\right]$$

$$= \frac{1}{n!}\sum_{\sigma_n \in S_n}\mathbb{E}_{(v,x,y)}\left[\sum_{i=1}^{n}\sum_{j=1}^{m} g_{\sigma_n^{-1}(i)j}(\sigma_n v, \sigma_n x, y)[\sigma_n v]_{\sigma_n^{-1}(i)j}\right] - \mathbb{E}_{(v,x,y)}\left[\sum_{i=1}^{n} p_i(v, x, y)\right]$$

$$= \frac{1}{n!}\sum_{\sigma_n \in S_n}\mathbb{E}_{(v,x,y)}\left[\sum_{i=1}^{n}\sum_{j=1}^{m} g_{ij}(\sigma_n v, \sigma_n x, y)[\sigma_n v]_{ij}\right] - \mathbb{E}_{(v,x,y)}\left[\sum_{i=1}^{n} p_i(v, x, y)\right]$$

$$= \frac{1}{n!}\sum_{\sigma_n \in S_n}\mathbb{E}_{(v,x,y)}\left[\sum_{i=1}^{n}\sum_{j=1}^{m} g_{ij}(v, x, y) v_{ij}\right] - \mathbb{E}_{(v,x,y)}\left[\sum_{i=1}^{n} p_i(v, x, y)\right]$$

$$= \mathbb{E}_{(v,x,y)}\left[\sum_{i=1}^{n}\sum_{j=1}^{m} g_{ij}(v, x, y) v_{ij}\right] - \mathbb{E}_{(v,x,y)}\left[\sum_{i=1}^{n} p_i(v, x, y)\right]$$

$$= \mathbb{E}_{(v,x,y)}\left[\sum_{i=1}^{n} u_i(v_i, v, x, y)\right].$$

Thus, we have completed the second step.

**Step 3:** We lastly prove that the auction mechanism has a smaller ex-post regret after projection, *i.e.*,

$$\mathbb{E}\left[\max_{v' \in \mathcal{V}^{n \times m}}\sum_{i=1}^{n}[\mathcal{Q}_1 u]_i(v_i, (v'_i, v_{-i}), x, y)\right] \le \mathbb{E}\left[\max_{v' \in \mathcal{V}^{n \times m}}\sum_{i=1}^{n} u(v_i, (v'_i, v_{-i}), x, y)\right].$$

For the simplicity, we denote $\sum_{j=1}^{m} g_{ij}v_{ij}$ by $\langle g_i, v_i \rangle$. Then, we have,

$$\mathbb{E}_{(v,x,y)}\left[\max_{v' \in \mathcal{V}^{n \times m}} \sum_{i=1}^{n} [\mathcal{Q}_1 u]_i\big(v_i, (v'_i, v_{-i}), x, y\big)\right]$$

$$= \mathbb{E}_{(v,x,y)}\left[\max_{v' \in \mathcal{V}^{n \times m}} \sum_{i=1}^{n} \sum_{j=1}^{m} [\mathcal{Q}_1 g]_{ij}\big((v'_i, v_{-i}), x, y\big)v_{ij} - [\mathcal{Q}_1 p]_i\big((v'_i, v_{-i}), x, y\big)\right]$$

$$= \mathbb{E}_{(v,x,y)}\left[\max_{v' \in \mathcal{V}^{n \times m}} \sum_{i=1}^{n} \big\langle [\mathcal{Q}_1 g]_i\big((v'_i, v_{-i}), x, y\big), v_i \big\rangle - [\mathcal{Q}_1 p]_i\big((v'_i, v_{-i}), x, y\big)\right]$$

$$= \mathbb{E}_{(v,x,y)}\left[\max_{v' \in \mathcal{V}^{n \times m}} \sum_{i=1}^{n} \frac{1}{n!} \sum_{\sigma_n \in S_n} \big\langle g_{\sigma_n^{-1}(i)}\big(\sigma_n(v'_i, v_{-i}), \sigma_n x, y\big), v_i \big\rangle - p_{\sigma_n^{-1}(i)}\big(\sigma_n(v'_i, v_{-i}), \sigma_n x, y\big)\right].$$

Combining the following inequality,

$$\max_{z} \sum_{k=1}^{K} f_k(z) \leq \sum_{k=1}^{K} \max_{z} f_k(z),$$

we have that

$$\mathbb{E}_{(v,x,y)}\left[\max_{v' \in \mathcal{V}^{n \times m}} \sum_{i=1}^{n} \frac{1}{n!} \sum_{\sigma_n \in S_n} \big\langle g_{\sigma_n^{-1}(i)}\big(\sigma_n(v'_i, v_{-i}), \sigma_n x, y\big), v_i \big\rangle - p_{\sigma_n^{-1}(i)}\big(\sigma_n(v'_i, v_{-i}), \sigma_n x, y\big)\right]$$

$$\leq \mathbb{E}_{(v,x,y)}\left[\frac{1}{n!} \sum_{\sigma_n \in S_n} \max_{v' \in \mathcal{V}^{n \times m}} \sum_{i=1}^{n} \big\langle g_{\sigma_n^{-1}(i)}\big(\sigma_n(v'_i, v_{-i}), \sigma_n x, y\big), v_i \big\rangle - p_{\sigma_n^{-1}(i)}\big(\sigma_n(v'_i, v_{-i}), \sigma_n x, y\big)\right]$$

$$= \frac{1}{n!} \sum_{\sigma_n \in S_n} \mathbb{E}_{(v,x,y)}\left[\max_{v' \in \mathcal{V}^{n \times m}} \sum_{i=1}^{n} \big\langle g_{\sigma_n^{-1}(i)}\big(\sigma_n(v'_i, v_{-i}), \sigma_n x, y\big), v_i \big\rangle - p_{\sigma_n^{-1}(i)}\big(\sigma_n(v'_i, v_{-i}), \sigma_n x, y\big)\right]$$

$$= \frac{1}{n!} \sum_{\sigma_n \in S_n} \mathbb{E}_{(v,x,y)}\left[\max_{v' \in \mathcal{V}^{n \times m}} \sum_{i=1}^{n} u_i\big(v_i, (v'_i, v_{-i}), x, y\big)\right]$$

$$= \mathbb{E}_{(v,x,y)}\left[\max_{v' \in \mathcal{V}^{n \times m}} \sum_{i=1}^{n} u_i\big(v_i, (v'_i, v_{-i}), x, y\big)\right].$$

We thus have completed the proof of eqs. (1) and (2) when the orbit averaging $\mathcal{Q}.$ is the condition of bidder-symmetry.

Then, we prove this theorem in the condition of item-symmetry; *i.e.*, the orbit averaging $\mathcal{Q}.$ is the item averaging $\mathcal{Q}_2$, acting on the allocation rule $g$ and the payment rule $p$ as shown below,

$$\mathcal{Q}_2 g(v, x, y) = \frac{1}{m!} \sum_{\sigma_m \in S_m} g(v\sigma_m, x, y\sigma_m)\sigma_m^{-1},$$

and,

$$\mathcal{Q}_2 p(v, x, y) = \frac{1}{m!} \sum_{\sigma_m \in S_m} p(v\sigma_m, x, y\sigma_m).$$

**Step 1:** We first prove that the auction mechanism has the same expected revenue after projection, *i.e.*,

$$\mathbb{E}_{(v,x,y)}\left[\sum_{i=1}^{n} [\mathcal{Q}_2 p]_i(v, x, y)\right] = \mathbb{E}_{(v,x,y)}\left[\sum_{i=1}^{n} p_i(v, x, y)\right].$$

Since the valuation joint distribution is invariant under bidder permutation, we have $\mathbb{E}_{(v,x,y)}[f(v\sigma_m, x, y\sigma_m)] = \mathbb{E}_{(v,x,y)}[f(v, x, y)]$. Then, we have

$$\mathbb{E}_{(v,x,y)}\left[\sum_{i=1}^{n}[\mathcal{Q}_2 p]_i(v, x, y)\right]$$

$$= \mathbb{E}_{(v,x,y)}\left[\frac{1}{m!}\sum_{i=1}^{n}\sum_{\sigma_m \in S_m} p_i(v\sigma_m, x, y\sigma_m)\right]$$

$$= \frac{1}{m!}\sum_{i=1}^{n}\sum_{\sigma_m \in S_m} \mathbb{E}_{(v,x,y)}\left[p_i(v\sigma_m, x, y\sigma_m)\right]$$

$$= \frac{1}{m!}\sum_{i=1}^{n}\sum_{\sigma_m \in S_m} \mathbb{E}_{(v,x,y)}\left[p_i(v, x, y)\right] = \sum_{i=1}^{n} \mathbb{E}_{(v,x,y)}\left[p_i(v, x, y)\right].$$

We have thus completed the first step.

**Step 2:** We then prove that the sum of all bidders' utilities remains same after projection, *i.e.*,

$$\mathbb{E}_{(v,x,y)}\left[\sum_{i=1}^{n}[\mathcal{Q}_2 u]_i(v_i, v, x, y)\right] = \mathbb{E}_{(v,x,y)}\left[\sum_{i=1}^{n} u_i(v_i, v, x, y)\right].$$

Given that for all permutation $\pi$,

$$\sum_{j=1}^{m} g_{i\pi(j)}v_{i\pi(j)} = \sum_{j=1}^{m} g_{ij}v_{ij},$$

we have the following equation,

$$\mathbb{E}_{(v,x,y)}\left[\sum_{i=1}^{n}[\mathcal{Q}_2 u]_i(v_i, v, x, y)\right]$$

$$= \mathbb{E}_{(v,x,y)}\left[\sum_{i=1}^{n}\left[\sum_{j=1}^{m}[\mathcal{Q}_2 g]_{ij}(v, x, y)v_{ij} - [\mathcal{Q}_2 p]_i(v, x, y)\right]\right]$$

$$= \mathbb{E}_{(v,x,y)}\left[\sum_{i=1}^{n}\sum_{j=1}^{m}[\mathcal{Q}_2 g]_{ij}(v, x, y)v_{ij}\right] - \mathbb{E}_{(v,x,y)}\left[\sum_{i=1}^{n} p_i(v, x, y)\right]$$

$$= \mathbb{E}_{(v,x,y)}\left[\frac{1}{m!}\sum_{i=1}^{n}\sum_{j=1}^{m}\sum_{\sigma_m \in S_m} g_{i\sigma_m^{-1}(j)}(v\sigma_m, x, y\sigma_m)v_{ij}\right] - \mathbb{E}_{(v,x,y)}\left[\sum_{i=1}^{n} p_i(v, x, y)\right]$$

$$= \frac{1}{m!}\sum_{\sigma_m \in S_m}\mathbb{E}_{(v,x,y)}\left[\sum_{i=1}^{n}\sum_{j=1}^{m} g_{i\sigma_m^{-1}(j)}(v\sigma_m, x, y\sigma_m)[v\sigma_m]_{i\sigma_m^{-1}(j)}\right] - \mathbb{E}_{(v,x,y)}\left[\sum_{i=1}^{n} p_i(v, x, y)\right]$$

$$= \frac{1}{m!}\sum_{\sigma_m \in S_m}\mathbb{E}_{(v,x,y)}\left[\sum_{i=1}^{n}\sum_{j=1}^{m} g_{ij}(v\sigma_m, x, y\sigma_m)[v\sigma_m]_{ij}\right] - \mathbb{E}_{(v,x,y)}\left[\sum_{i=1}^{n} p_i(v, x, y)\right]$$

$$= \frac{1}{m!}\sum_{\sigma_m \in S_m}\mathbb{E}_{(v,x,y)}\left[\sum_{i=1}^{n}\sum_{j=1}^{m} g_{ij}(v, x, y)v_{ij}\right] - \mathbb{E}_{(v,x,y)}\left[\sum_{i=1}^{n} p_i(v, x, y)\right]$$

$$= \mathbb{E}_{(v,x,y)}\left[\sum_{i=1}^{n}\sum_{j=1}^{m} g_{ij}(v, x, y)v_{ij}\right] - \mathbb{E}_{(v,x,y)}\left[\sum_{i=1}^{n} p_i(v, x, y)\right]$$

$$= \mathbb{E}_{(v,x,y)}\left[\sum_{i=1}^{n} u_i(v_i, v, x, y)\right].$$

Thus, we have completed the second step.

**Step 3:** We lastly prove that the auction mechanism have a smaller ex-post regret after projection, *i.e.*,

$$\mathbb{E}\left[\max_{v' \in \mathcal{V}^{n \times m}} \sum_{i=1}^{n} [\mathcal{Q}_2 u]_i(v_i, (v'_i, v_{-i}), x, y)\right] \leq \mathbb{E}\left[\max_{v' \in \mathcal{V}^{n \times m}} \sum_{i=1}^{n} u(v_i, (v'_i, v_{-i}), x, y)\right].$$

By the definition of the utility $u$ and the item averaging $\mathcal{Q}_2$, we have

$$\mathop{\mathbb{E}}_{(v,x,y)}\left[\max_{v' \in \mathcal{V}^{n \times m}} \sum_{i=1}^{n} [\mathcal{Q}_2 u]_i\big(v_i, (v'_i, v_{-i}), x, y\big)\right]$$

$$= \mathop{\mathbb{E}}_{(v,x,y)}\left[\max_{v' \in \mathcal{V}^{n \times m}} \sum_{i=1}^{n} \sum_{j=1}^{m} [\mathcal{Q}_2 g]_{ij}\big((v'_i, v_{-i}), x, y\big) v_{ij} - [\mathcal{Q}_2 p]_i\big((v'_i, v_{-i}), x, y\big)\right]$$

$$= \mathop{\mathbb{E}}_{(v,x,y)}\left[\max_{v' \in \mathcal{V}^{n \times m}} \sum_{i=1}^{n} \big\langle [\mathcal{Q}_2 g]_i\big((v'_i, v_{-i}), x, y\big), v_i \big\rangle - [\mathcal{Q}_2 p]_i\big((v'_i, v_{-i}), x, y\big)\right]$$

$$= \mathop{\mathbb{E}}_{(v,x,y)}\left[\max_{v' \in \mathcal{V}^{n \times m}} \sum_{i=1}^{n} \frac{1}{m!} \sum_{\sigma_m \in S_m} \big\langle g_i\big((v'_i, v_{-i})\sigma_m, x, y\sigma_m\big)\sigma_m^{-1}, v_i \big\rangle - p_i\big((v'_i, v_{-i})\sigma_m, x, y\sigma_m\big)\right].$$

Combining the following inequality

$$\max_z \sum_{k=1}^{K} f_k(z) \leq \sum_{k=1}^{K} \max_z f_k(z),$$

we have

$$\mathop{\mathbb{E}}_{(v,x,y)}\left[\max_{v' \in \mathcal{V}^{n \times m}} \sum_{i=1}^{n} \frac{1}{m!} \sum_{\sigma_m \in S_m} \big\langle g_i\big((v'_i, v_{-i})\sigma_m, x, y\sigma_m\big)\sigma_m^{-1}, v_i \big\rangle - p_i\big((v'_i, v_{-i})\sigma_m, x, y\sigma_m\big)\right]$$

$$\leq \mathop{\mathbb{E}}_{(v,x,y)}\left[\frac{1}{m!} \sum_{\sigma_m \in S_m} \sum_{i=1}^{n} \max_{v' \in \mathcal{V}^{n \times m}} \big\langle g_i\big((v'_i, v_{-i})\sigma_m, x, y\sigma_m\big)\sigma_m^{-1}, v_i \big\rangle - p_i\big((v'_i, v_{-i})\sigma_m, x, y\sigma_m\big)\right]$$

$$= \frac{1}{m!} \sum_{\sigma_m \in S_m} \mathop{\mathbb{E}}_{(v,x,y)}\left[\max_{v' \in \mathcal{V}^{n \times m}} \sum_{i=1}^{n} \big\langle g_i\big((v'_i, v_{-i})\sigma_m, x, y\sigma_m\big), v_i\sigma_m \big\rangle - p_i\big((v'_i, v_{-i})\sigma_m, x, y\sigma_m\big)\right]$$

$$= \frac{1}{m!} \sum_{\sigma_m \in S_m} \mathop{\mathbb{E}}_{(v,x,y)}\left[\max_{v' \in \mathcal{V}^{n \times m}} \sum_{i=1}^{n} u_i\big(v_i, (v'_i, v_{-i}), x, y\big)\right]$$

$$= \mathop{\mathbb{E}}_{(v,x,y)}\left[\max_{v' \in \mathcal{V}^{n \times m}} \sum_{i=1}^{n} u_i\big(v_i, (v'_i, v_{-i}), x, y\big)\right].$$

We thus have proved this theorem in the condition of item-symmetry.

The proofs are completed. $\hfill\square$

### A.3 Proof of Lemma 3.3

In this section, we present the proof of Lemma 3.3.

*Proof of Lemma 3.3.* Both the bidder averaging and the item averaging are linear. Thus, we have the following results,

$$\mathcal{Q}_1 \circ \mathcal{Q}_2 f(v, x, y)$$

$$= \mathcal{Q}_1 \left[ \frac{1}{m!} \sum_{\sigma_m \in S_m} f(v\sigma_m, x, y\sigma_m)\sigma_m^{-1} \right]$$

$$= \frac{1}{n!} \sum_{\sigma_n \in S_n} \sigma_n^{-1} \left[ \frac{1}{m!} \sum_{\sigma_m \in S_m} f(\sigma_n v\sigma_m, \sigma_n x, y\sigma_m)\sigma_m^{-1} \right]$$

$$= \frac{1}{n!m!} \sum_{\sigma_n \in S_n} \sum_{\sigma_m \in S_m} \sigma_n^{-1} f(\sigma_n v\sigma_m, \sigma_n x, y\sigma_m)\sigma_m^{-1} = \mathcal{Q}_3 f(v, x, y),$$

and

$$\mathcal{Q}_2 \circ \mathcal{Q}_1 f(v, x, y)$$

$$= \mathcal{Q}_2 \left[ \frac{1}{n!} \sum_{\sigma_n \in S_n} \sigma_n^{-1} f(\sigma_n v, \sigma_n x, y) \right]$$

$$= \frac{1}{m!} \sum_{\sigma_m \in S_m} \left[ \frac{1}{n!} \sum_{\sigma_n \in S_n} \sigma_n^{-1} f(\sigma_n v\sigma_m, \sigma_n x, y\sigma_m) \right] \sigma_m^{-1}$$

$$= \frac{1}{m!n!} \sum_{\sigma_n \in S_n} \sum_{\sigma_m \in S_m} \sigma_n^{-1} f(\sigma_n v\sigma_m, \sigma_n x, y\sigma_m)\sigma_m^{-1} = \mathcal{Q}_3 f(v, x, y).$$

The above two equations hold for any $f$. Then, we may prove that

$$\mathcal{Q}_3 = \mathcal{Q}_1 \circ \mathcal{Q}_2 = \mathcal{Q}_2 \circ \mathcal{Q}_1,$$

which is exactly the claim of this theorem.

The proof is completed. □

### A.4 Proof of Theorem 3.4

In this section, we apply our Lemma 3.3 and Theorem 3.1 to prove Theorem 3.4.

*Proof of Theorem 3.4.* For the simplicity, we rewrite $\mathcal{Q}_3 p$ and $\mathcal{Q}_3 reg$ as $\mathcal{Q}_2(\mathcal{Q}_1 p)$ and $\mathcal{Q}_2(\mathcal{Q}_1 reg)$, respectively. Then, for a payment rule $p$, we have that,

$$\mathbb{E}_{(v,x,y)} \left[ \sum_{i=1}^{n} [\mathcal{Q}_3 p]_i(v, x, y) \right]$$

$$= \mathbb{E}_{(v,x,y)} \left[ \sum_{i=1}^{n} [\mathcal{Q}_2(\mathcal{Q}_1 p)]_i(v, x, y) \right] = \mathbb{E}_{(v,x,y)} \left[ \sum_{i=1}^{n} [\mathcal{Q}_1 p]_i(v, x, y) \right]$$

$$= \mathbb{E}_{(v,x,y)} \left[ \sum_{i=1}^{n} p_i(v, x, y) \right].$$

Also, we have the following result,

$$\mathbb{E}_{(v,x,y)} \left[ \sum_{i=1}^{n} [\mathcal{Q}_3 reg]_i(v, x, y) \right]$$

$$= \mathbb{E}_{(v,x,y)} \left[ \sum_{i=1}^{n} [\mathcal{Q}_2(\mathcal{Q}_1 reg)]_i(v, x, y) \right] \leq \mathbb{E}_{(v,x,y)} \left[ \sum_{i=1}^{n} [\mathcal{Q}_1 reg]_i(v, x, y) \right]$$

$$\leq \mathbb{E}_{(v,x,y)} \left[ \sum_{i=1}^{n} reg_i(v, x, y) \right].$$

Moreover, we have the following result on the regret gap $\Delta_3$,

$$\mathop{\mathbb{E}}_{(v,x,y)}\left[\Delta_3(g,p;v,x,y)\right] = \mathop{\mathbb{E}}_{(v,x,y)}\left[\Delta_1(g,p;v,x,y)\right] + \mathop{\mathbb{E}}_{(v,x,y)}\left[\Delta_2(\mathcal{Q}_1 g, \mathcal{Q}_1 p; v,x,y)\right] \geq 0.$$

This proof is completed. $\qquad\qquad\qquad\qquad\qquad\qquad\qquad\qquad\qquad\qquad\qquad\qquad\qquad\quad$ $\square$

## A.5 Proof of Theorem 3.5

In this section, we present the proof of Theorem 3.5.

We start with the definitions or notations necessary for our proof. We define the allocation rule space and the payment rule space as follows,

$$\mathcal{G} = \{g^\omega : \omega \in \Omega\} \text{ and } \mathcal{P} = \{p^\omega : \omega \in \Omega\},$$

where $\omega$ is the auction mechanism parameter and $\Omega$ is the set of all feasible parameters. We then define the induced utility and ex-post regret spaces as follows,

$$\mathcal{U} = \left\{ u^\omega : u_i^\omega(v_i', v, x, y) = \sum_{j=1}^m g_{ij}^\omega(v,x,y)v_{ij}' - p_i^\omega(v,x,y) \right\},$$

and

$$\mathcal{R} = \left\{ reg^\omega : reg_i^\omega(v,x,y) = \max_{v_i'} u^\omega(v_i, (v_i', v_{-i}), x, y) - u^\omega(v_i, v, x, y) \right\}.$$

Then, the $l_{\infty,1}$-distance on $\mathcal{U}$ and $\mathcal{P}$ is defined as below,

$$l_{\infty,1}(u, u') = \max_{(v,v_i',x,y)} \left( \sum_{i=1}^n |u_i(v_i, (v_i', v_{-i}), x, y) - u_i'(v_i, (v_i', v_{-i}), x, y)| \right),$$

and

$$l_{\infty,1}(p, p') = \max_{(v,x,y)} \left( \sum_{i=1}^n |p_i(v,x,y) - p_i'(v,x,y)| \right).$$

We now present the proof of Theorem 3.5.

*Proof of Theorem 3.5.* We prove Theorem 3.5 in two steps: (1) we first prove that the distance between any two mechanisms is smaller when we project them to be permutation-equivariant; (2) then, we prove that the smaller distance implies a smaller covering number.

**Step 1:** We prove that the distance between two mechanisms becomes smaller after projection, *i.e.*,

$$l_{\infty,1}(\mathcal{Q}.p, \mathcal{Q}.p') \leq l_{\infty,1}(p, p'),$$

and

$$l_{\infty,1}(\mathcal{Q}.u, \mathcal{Q}.u') \leq l_{\infty,1}(u, u'),$$

where $u, u' \in \mathcal{U}$, $p, p' \in \mathcal{P}$, and $\mathcal{Q}. = \mathcal{Q}_1$ or $\mathcal{Q}_2$.

When $\mathcal{Q}.$ is $\mathcal{Q}_1$, we have that

$$l_{\infty,1}(\mathcal{Q}_1 p, \mathcal{Q}_1 p')$$

$$= \max_{(v,x,y)} \sum_{i=1}^{n} |\mathcal{Q}_1 p_i(v,x,y) - \mathcal{Q}_1 p_i'(v,x,y)|$$

$$= \max_{(v,x,y)} \sum_{i=1}^{n} \left| \frac{1}{n!} \sum_{\sigma_n \in S_n} \left[ p_{\sigma_n^{-1}(i)}(\sigma_n v, \sigma_n x, y) - p_{\sigma_n^{-1}(i)}'(\sigma_n v, \sigma_n x, y) \right] \right|$$

$$\leq \max_{(v,x,y)} \sum_{i=1}^{n} \frac{1}{n!} \sum_{\sigma_n \in S_n} \left| p_{\sigma_n^{-1}(i)}(\sigma_n v, \sigma_n x, y) - p_{\sigma_n^{-1}(i)}'(\sigma_n v, \sigma_n x, y) \right|$$

$$\leq \sum_{\sigma_n \in S_n} \frac{1}{n!} \max_{(v,x,y)} \sum_{i=1}^{n} \left| p_{\sigma_n^{-1}(i)}(\sigma_n v, \sigma_n x, y) - p_{\sigma_n^{-1}(i)}'(\sigma_n v, \sigma_n x, y) \right|$$

$$= \sum_{\sigma_n \in S_n} \frac{1}{n!} \max_{(v,x,y)} \sum_{i=1}^{n} \left| p_i(\sigma_n v, \sigma_n x, y) - p_i'(\sigma_n v, \sigma_n x, y) \right|$$

$$= \sum_{\sigma_n \in S_n} \frac{1}{n!} \max_{(v,x,y)} \sum_{i=1}^{n} \left| p_i(v,x,y) - p_i'(v,x,y) \right|$$

$$= \max_{(v,x,y)} \sum_{i=1}^{n} \left| p_i(v,x,y) - p_i'(v,x,y) \right|$$

$$= l_{\infty,1}(p, p'),$$

and

$$l_{\infty,1}(\mathcal{Q}_1 u, \mathcal{Q}_1 u')$$

$$= \max_{v,v',x,y} \sum_{i=1}^{n} |[\mathcal{Q}_1 u]_i(v_i, (v_i', v_{-i}), x, y) - [\mathcal{Q}_1 u']_i(v_i, (v_i', v_{-i}), x, y)|$$

$$= \max_{v,v',x,y} \sum_{i=1}^{n} \left| \frac{1}{n!} \sum_{\sigma_n \in S_n} u_{\sigma_n^{-1}(i)}(v_i, \sigma_n(v_i', v_{-i}), \sigma_n x, y) - u_{\sigma_n^{-1}(i)}'(v_i, \sigma_n(v_i', v_{-i}), \sigma_n x, y) \right|$$

$$\leq \max_{v,v',x,y} \sum_{i=1}^{n} \frac{1}{n!} \sum_{\sigma_n \in S_n} |u_{\sigma_n^{-1}(i)}(v_i, \sigma_n(v_i', v_{-i}), \sigma_n x, y) - u_{\sigma_n^{-1}(i)}'(v_i, \sigma_n(v_i', v_{-i}), \sigma_n x, y)|$$

$$\leq \frac{1}{n!} \sum_{\sigma_n \in S_n} \max_{v,v',x,y} \sum_{i=1}^{n} |u_{\sigma_n^{-1}(i)}(v_i, \sigma_n(v_i', v_{-i}), \sigma_n x, y) - u_{\sigma_n^{-1}(i)}'(v_i, \sigma_n(v_i', v_{-i}), \sigma_n x, y)|$$

$$= \frac{1}{n!} \sum_{\sigma_n \in S_n} \max_{v,v',x,y} \sum_{i=1}^{n} |u_i(v_i, (v_i', v_{-i}), x, y) - u_i'(v_i, (v_i', v_{-i}), x, y)|$$

$$= \max_{v,v',x,y} \sum_{i=1}^{n} |u_i(v_i, (v_i', v_{-i}), x, y) - u_i'(v_i, (v_i', v_{-i}), x, y)|$$

$$= l_{\infty,1}(u, u').$$

Then, when $\mathcal{Q}.$ is $\mathcal{Q}_2$, we prove the result as below,

$$l_{\infty,1}(\mathcal{Q}_2 p, \mathcal{Q}_2 p')$$

$$= \max_{(v,x,y)} \sum_{i=1}^{n} |\mathcal{Q}_2 p_i(v,x,y) - \mathcal{Q}_2 p_i'(v,x,y)|$$

$$= \max_{(v,x,y)} \sum_{i=1}^{n} \left| \frac{1}{m!} \sum_{\sigma_m \in S_m} \left[ p_i(v\sigma_m, x, y\sigma_m) - p_i'(v\sigma_m, x, y\sigma_m) \right] \right|$$

$$\leq \max_{(v,x,y)} \sum_{i=1}^{n} \frac{1}{m!} \sum_{\sigma_m \in S_m} \left| p_i(v\sigma_m, x, y\sigma_m) - p_i'(v\sigma_m, x, y\sigma_m) \right|$$

$$\leq \sum_{\sigma_m \in S_m} \frac{1}{m!} \max_{(v,x,y)} \sum_{i=1}^{n} \left| p_i(v\sigma_m, x, y\sigma_m) - p_i'(v\sigma_m, x, y\sigma_m) \right|$$

$$= \sum_{\sigma_m \in S_m} \frac{1}{m!} \max_{(v,x,y)} \sum_{i=1}^{n} \left| p_i(v, x, y) - p_i'(v, x, y) \right|$$

$$= \max_{(v,x,y)} \sum_{i=1}^{n} \left| p_i(v, x, y) - p_i'(v, x, y) \right|$$

$$= l_{\infty,1}(p, p'),$$

and

$$l_{\infty,1}(\mathcal{Q}_2 u, \mathcal{Q}_2 u')$$

$$= \max_{v,v',x,y} \sum_{i=1}^{n} |[\mathcal{Q}_2 u]_i(v_i, (v_i', v_{-i}), x, y) - [\mathcal{Q}_2 u']_i(v_i, (v_i', v_{-i}), x, y)|$$

$$= \max_{v,v',x,y} \sum_{i=1}^{n} \left| \frac{1}{m!} \sum_{\sigma_m \in S_m} |u_i(v_i \sigma_m, (v_i', v_{-i})\sigma_m, x, y\sigma_m) - u_i'(v_i \sigma_m, (v_i', v_{-i})\sigma_m, x, y\sigma_m)| \right|$$

$$\leq \max_{v,v',x,y} \sum_{i=1}^{n} \frac{1}{m!} \sum_{\sigma_m \in S_m} |u_i(v_i \sigma_m, (v_i', v_{-i})\sigma_m, x, y\sigma_m) - u_i'(v_i \sigma_m, (v_i', v_{-i})\sigma_m, x, y\sigma_m)|$$

$$\leq \frac{1}{m!} \sum_{\sigma_m \in S_m} \max_{v,v',x,y} \sum_{i=1}^{n} |u_i(v_i \sigma_m, (v_i', v_{-i})\sigma_m, x, y\sigma_m) - u_i'(v_i \sigma_m, (v_i', v_{-i})\sigma_m, x, y\sigma_m)|$$

$$= \frac{1}{m!} \sum_{\sigma_m \in S_m} \max_{v,v',x,y} \sum_{i=1}^{n} |u_i(v_i, (v_i', v_{-i}), x, y) - u_i'(v_i, (v_i', v_{-i}), x, y)|$$

$$= \max_{v,v',x,y} \sum_{i=1}^{n} |u_i(v_i, (v_i', v_{-i}), x, y) - u_i'(v_i, (v_i', v_{-i}), x, y)|$$

$$= l_{\infty,1}(u, u').$$

Thus, we have completed Step 1.

**Step 2:** We prove that a smaller distance implies a smaller covering number.

Let $\mathcal{X}$ and $\mathcal{Y}$ be two metric spaces with two different distances $l_1$ and $l_2$, respectively. There exists a surjective mapping $f$ from $\mathcal{Y}$ to $\mathcal{X}$, such that $l_1(f(x), f(y)) \leq l_2(x, y)$ for all $x, y \in \mathcal{Y}$. The covering numbers $\mathcal{N}_1(\mathcal{X}, r)$ and $\mathcal{N}_2(\mathcal{Y}, r)$ are defined as the minimum numbers of balls with radius r that can cover $\mathcal{X}$ and $\mathcal{Y}$ under $l_1$ and $l_2$, respectively.

By the definition of the covering number $\mathcal{N}_2(\mathcal{Y}, r)$, there exists a set $\mathcal{A}$ of scale $\mathcal{N}_2(\mathcal{Y}, r)$, such that

$$l_2(x, \mathcal{A}) = \inf_{y \in \mathcal{A}} l_2(x, y) < r, \forall x \in \mathcal{Y}.$$

Then, $f(\mathcal{A})$ is also a $r$-cover for $\mathcal{X}$ under distance $l_1$, *i.e.*, for any $x \in \mathcal{Y}$, we have

$$l_1(f(x), f(\mathcal{A})) = \inf_{y \in \mathcal{A}} l_1(f(x), f(y)) \leq \inf_{y \in \mathcal{A}} l_2(x, y) = l_2(x, \mathcal{A}) < r.$$

Because $f$ is surjective, for any $x' \in \mathcal{X}$, there exists an $x \in \mathcal{Y}$, such that $x' = f(x)$. Then, for any $x' \in \mathcal{X}$, we have that

$$l_1(x', f(\mathcal{A})) = l_1(f(x), f(\mathcal{A})) < r.$$

By the definition of $\mathcal{N}_1(\mathcal{X}, r)$, we have

$$\mathcal{N}_1(\mathcal{X}, r) \leq |f(\mathcal{A})| \leq |\mathcal{A}| = \mathcal{N}_2(\mathcal{Y}, r).$$

Eventually, combining the results in Step 1 and in Step 2, we have that

$$\mathcal{N}_{\infty,1}(\mathcal{Q}.\mathcal{U}, r) \leq \mathcal{N}_{\infty,1}(\mathcal{U}, r),$$

and

$$\mathcal{N}_{\infty,1}(\mathcal{Q}.\mathcal{P}, r) \leq \mathcal{N}_{\infty,1}(\mathcal{P}, r),$$

for both the bidder averaging $\mathcal{Q}_1$ and the item averaging $\mathcal{Q}_2$. $\qquad\square$

## A.6 Proof of Theorem 3.6 and Corollary 3.7

We first introduce two lemmas. The first lemma gives a concentration inequality via the covering number. This result can be used to bound the gap between expected revenue/ex-post regret and empirical revenue/ex-post regret. The second lemma bounds the covering number $\mathcal{N}_{\infty,1}(\mathcal{R}, 2r)$ by the covering number $\mathcal{N}_{\infty,1}(\mathcal{U}, r)$. Both lemmas has been proved by [10]. We recall them here to make our paper completed.

**Lemma A.1** (cf. Lemma E.1, [10]). *Let $\mathcal{S} = \{z_1, \ldots, z_L\}$ be a set of i.i.d. sample points drawn from a distribution $\mathcal{D}$ over $\mathcal{Z}$. Suppose $\mathcal{F}$ is a set of functions from $\mathcal{Z}$ to $\mathbb{R}$ such that $f(z) \in [a, b]$ for all $f \in \mathcal{F}$ and $z \in \mathcal{Z}$. We define $l_\infty$ on $\mathcal{F}$ as*

$$l_\infty(f, f') = \max_{z \in \mathcal{Z}} |f(z) - f'(z)|,$$

*and $\mathcal{N}_\infty(\mathcal{F}, r)$ as the minimum number of balls with radius $r$ that can cover $\mathcal{F}$ under $l_\infty$-distance. Then, we have the following concentration inequality,*

$$\mathbb{P}\left[\exists f \in \mathcal{F} : \left| \frac{1}{L} \sum_{i=1}^{L} f(z_i) - \mathbb{E}[f(z)] \right| > \epsilon \right] \leq 2\mathcal{N}_\infty\left(\mathcal{F}, \frac{\epsilon}{3}\right) \exp\left(-\frac{2L\epsilon^2}{9(b-a)^2}\right).$$

*Proof.* By the definition of $\mathcal{N}_\infty(\mathcal{F}, r)$, for any $f \in \mathcal{F}$, there exists an $f_r \in \mathcal{F}_r$ such that $\mathcal{F}_r$ is an $r$-cover for $\mathcal{F}$ and $l_\infty(f, f_r) < r$. Denote $\frac{1}{L} \sum_{i=1}^{L} f(z_i)$ by $\mathbb{E}_{\mathcal{S}}[f(z)]$. Then, we have

$$\mathbb{P}\left[\exists f \in \mathcal{F} : \left| \mathbb{E}_{\mathcal{S}}[f(z)] - \mathbb{E}[f(z)] \right| > \epsilon \right]$$

$$= \mathbb{P}\left[\exists f \in \mathcal{F} : \left| \mathbb{E}_{\mathcal{S}}[f(z)] - \mathbb{E}_{\mathcal{S}}[f_r(z)] + \mathbb{E}_{\mathcal{S}}[f_r(z)] - \mathbb{E}[f_r(z)] + \mathbb{E}[f_r(z)] - \mathbb{E}[f(z)] \right| > \epsilon \right]$$

$$\leq \mathbb{P}\left[\exists f \in \mathcal{F} : \left| \mathbb{E}_{\mathcal{S}}[f(z)] - \mathbb{E}_{\mathcal{S}}[f_r(z)] \right| + \left| \mathbb{E}_{\mathcal{S}}[f_r(z)] - \mathbb{E}[f_r(z)] \right| + \left| \mathbb{E}[f_r(z)] - \mathbb{E}[f(z)] \right| > \epsilon \right]$$

$$\leq \mathbb{P}\left[\exists f_r \in \mathcal{F}_{\frac{\epsilon}{3}} : \left| \mathbb{E}_{\mathcal{S}}[f_r(z)] - \mathbb{E}[f_r(z)] \right| > \frac{\epsilon}{3} \right]$$

$$\leq \mathcal{N}_\infty\left(\mathcal{F}, \frac{\epsilon}{3}\right) \mathbb{P}\left[\left| \mathbb{E}_{\mathcal{S}}[f(z)] - \mathbb{E}[f(z)] \right| > \frac{\epsilon}{3} \right]$$

$$\leq 2\mathcal{N}_\infty\left(\mathcal{F}, \frac{\epsilon}{3}\right) \exp\left(-\frac{2L\epsilon^2}{9(b-a)^2}\right).$$

The third inequality follows from the fact that when $r = \frac{\epsilon}{3}$, we have

$$|f(z) - f_r(z)| < \frac{\epsilon}{3},$$

for all $z \in \mathcal{Z}$ and $f \in \mathcal{F}$. Then, from the Hoeffding's inequality, we have

$$\left| \frac{1}{L} \sum_{i=1}^{L} f(z_i) - \frac{1}{L} \sum_{i=1}^{L} f_r(z_i) \right| < \frac{\epsilon}{3} \quad \text{and} \quad \left| \mathbb{E}[f(z)] - \mathbb{E}[f_r(z)] \right| < \frac{\epsilon}{3}.$$

The proof is completed. $\qquad\square$

The following lemma bounds the covering number $\mathcal{N}_{\infty,1}(\mathcal{R}, 2r)$ by the covering number $\mathcal{N}_{\infty,1}(\mathcal{U}, r)$. Then the gap between expected ex-post regret and empirical ex-post regret can be bounded by the covering number $\mathcal{N}_{\infty,1}(\mathcal{U}, r)$.

**Lemma A.2** (cf. Lemma E.3, [10]). *We define $\mathcal{N}_{\infty,1}(\mathcal{R}, r)$ and $\mathcal{N}_{\infty,1}(\mathcal{U}, r)$ as the minimum numbers of balls with radius $r$ that can cover spaces $\mathcal{R}$ and $\mathcal{U}$ under distance $l_{\infty,1}$, respectively. Then, we have that*

$$\mathcal{N}_{\infty,1}(\mathcal{R}, 2r) \leq \mathcal{N}_{\infty,1}(\mathcal{U}, r)$$

*Proof.* By the definition of $\mathcal{N}_{\infty,1}(\mathcal{U}, r)$, there exists an $r$-cover $\mathcal{U}_r$ for $\mathcal{U}$, such that $|\mathcal{U}_r| = \mathcal{N}_{\infty,1}(\mathcal{U}, r)$ and for any $u \in \mathcal{U}$,

$$l_{\infty,1}(u, \mathcal{U}_r) = \inf_{u' \in \mathcal{U}_r} l_{\infty,1}(u, u') < r.$$

We define $\mathcal{R}_r$ as

$$\{reg \in \mathcal{R} : reg_i(v, x, y) = \max_{v_i'} u_i(v_i, (v_i', v_{-i}), x, y) - u_i(v_i, v, x, y) \text{ for some } u_i \in \mathcal{U}_r\}.$$

Then, we can prove that $\mathcal{R}_r$ is a $2r$-cover for the space $\mathcal{R}$, *i.e.*,

$$l_{\infty,1}(reg, \mathcal{R}_r)$$
$$= \inf_{reg' \in \mathcal{R}_r} l_{\infty,1}(reg, reg')$$
$$= \inf_{reg' \in \mathcal{R}_r} \max_{v,x,y} \sum_{i=1}^{n} |reg_i(v, x, y) - reg_i'(v, x, y)|$$
$$= \inf_{u' \in \mathcal{U}_r} \max_{v,x,y} \sum_{i=1}^{n} \left| [\max_{v_i'} u_i(v_i, (v_i', v_{-i}), x, y) - u_i(v_i, v, x, y)] \right.$$
$$\left. - [\max_{v_i'} u_i'(v_i, (v_i', v_{-i}), x, y) - u_i'(v_i, v, x, y)] \right|$$
$$\leq \inf_{u' \in \mathcal{U}_r} \max_{v,x,y} \left[ \sum_{i=1}^{n} \left| \max_{v_i'} u_i(v_i, (v_i', v_{-i}), x, y) - \max_{v_i'} u_i'(v_i, (v_i', v_{-i}), x, y) \right| \right.$$
$$\left. + \left| u_i(v_i, v, x, y) - u_i'(v_i, v, x, y) \right| \right]$$
$$\leq \max_{v,x,y} \left[ \sum_{i=1}^{n} \left| \max_{v_i'} u_i(v_i, (v_i', v_{-i}), x, y) - \max_{v_i'} u_i^*(v_i, (v_i', v_{-i}), x, y) \right| \right.$$
$$\left. + \left| u_i(v_i, v, x, y) - u_i^*(v_i, v, x, y) \right| \right] \quad \text{(where } l_{\infty,1}(u, u^*) < r)$$
$$\leq \max_{v,x,y} \sum_{i=1}^{n} \left| \max_{v_i'} u_i(v_i, (v_i', v_{-i}), x, y) - \max_{v_i'} u_i^*(v_i, (v_i', v_{-i}), x, y) \right| + r$$
$$\leq \max_{v,x,y} \sum_{i=1}^{n} \max_{v_i'} |u_i(v_i, (v_i', v_{-i}), x, y) - u_i^*(v_i, (v_i', v_{-i}), x, y)| + r$$
$$= \max_{v,v_i',x,y} \sum_{i=1}^{n} |u_i(v_i, (v_i', v_{-i}), x, y) - u_i^*(v_i, (v_i', v_{-i}), x, y)| + r < 2r.$$

Eventually, we have

$$\mathcal{N}_{\infty,1}(\mathcal{R}, 2r) \leq |\mathcal{R}_r| \leq |\mathcal{U}_r| = \mathcal{N}_{\infty,1}(\mathcal{U}, r).$$

The proof is completed. □

We now prove Theorem 3.6 and Corollary 3.7.

*Proof of Theorem 3.6 and Corollary 3.7.* Applying Lemma A.1 to the spaces $\mathcal{P}$ and $\mathcal{U}$, we have that

$$\mathbb{P}\left[\exists \omega \in \Omega : \left| \mathbb{E}_{(v,x,y)}\left[\sum_{i=1}^{n} p_i^{\omega}(v,x,y)\right] - \frac{1}{L}\sum_{l=1}^{L}\sum_{i=1}^{n} p_i^{\omega}(v^{(l)},x^{(l)},y^{(l)}) \right| > \epsilon \right]$$

$$\leq 2\mathcal{N}_{\infty,1}\left(\mathcal{P}, \frac{\epsilon}{3}\right)\exp\left(-\frac{2L\epsilon^2}{9n^2}\right),$$

and

$$\mathbb{P}\left[\exists \omega \in \Omega : \left| \mathbb{E}_{(v,x,y)}\left[\sum_{i=1}^{n} reg_i^{\omega}(v,x,y)\right] - \frac{1}{L}\sum_{l=1}^{L}\sum_{i=1}^{n} reg_i^{\omega}(v^{(l)},x^{(l)},y^{(l)}) \right| > \epsilon \right]$$

$$\leq 2\mathcal{N}_{\infty,1}\left(\mathcal{R}, \frac{\epsilon}{3}\right)\exp\left(-\frac{2L\epsilon^2}{9n^2}\right)$$

$$\leq 2\mathcal{N}_{\infty,1}\left(\mathcal{U}, \frac{\epsilon}{6}\right)\exp\left(-\frac{2L\epsilon^2}{9n^2}\right),$$

where the last inequality follows from Lemma A.2.

Further, we assume that

$$\epsilon \geq \sqrt{\frac{9n^2}{2L}\left(\log\frac{4}{\delta} + \max\left\{\log\mathcal{N}_{\infty,1}\left(\mathcal{P},\frac{\epsilon}{3}\right), log\mathcal{N}_{\infty,1}\left(\mathcal{U},\frac{\epsilon}{6}\right)\right\}\right)}.$$

Then, we have the following inequalities,

$$\mathbb{P}\left[\exists \omega \in \Omega : \left| \mathbb{E}_{(v,x,y)}\left[\sum_{i=1}^{n} p_i^{\omega}(v,x,y)\right] - \frac{1}{L}\sum_{l=1}^{L}\sum_{i=1}^{n} p_i^{\omega}(v^{(l)},x^{(l)},y^{(l)}) \right| > \epsilon \right] \leq \frac{\delta}{2},$$

and

$$\mathbb{P}\left[\exists \omega \in \Omega : \left| \mathbb{E}_{(v,x,y)}\left[\sum_{i=1}^{n} reg_i^{\omega}(v,x,y)\right] - \frac{1}{L}\sum_{l=1}^{L}\sum_{i=1}^{n} reg_i^{\omega}(v^{(l)},x^{(l)},y^{(l)}) \right| > \epsilon \right] \leq \frac{\delta}{2}.$$

Thus, with probability at least $1 - \delta$, for any $\omega \in \Omega$, we have that

$$\left| \mathbb{E}_{(v,x,y)}\left[\sum_{i=1}^{n} p_i^{\omega}(v,x,y)\right] - \frac{1}{L}\sum_{l=1}^{L}\sum_{i=1}^{n} p_i^{\omega}(v^{(l)},x^{(l)},y^{(l)}) \right| < \epsilon, \tag{5}$$

and

$$\left| \mathbb{E}_{(v,x,y)}\left[\sum_{i=1}^{n} reg_i^{\omega}(v,x,y)\right] - \frac{1}{L}\sum_{l=1}^{L}\sum_{i=1}^{n} reg_i^{\omega}(v^{(l)},x^{(l)},y^{(l)}) \right| < \epsilon. \tag{6}$$

Equivalently, when the number of samples $L$ is large enough, *i.e.*,

$$L \geq \frac{9n^2}{2\epsilon^2}\left(\log\frac{4}{\delta} + \max\left\{\log\mathcal{N}_{\infty,1}\left(\mathcal{P},\frac{\epsilon}{3}\right), log\mathcal{N}_{\infty,1}\left(\mathcal{U},\frac{\epsilon}{6}\right)\right\}\right),$$

then, the eqs. (5) and (6) both hold with probability at least $1 - \delta$.

The proof is completed. $\qquad\square$

## A.7 Proof of the Generalization Bound for Myerson Auctions

Denote $rev(v,x,y)$ as $\sum_{i=1}^{n} p_i(v,x,y)$, then we have the following theorem,

**Theorem A.3.** *Assume the item valuation for each bidder is not larger than 1. When the sample complexity satisfies $L \geq \frac{1}{2\epsilon^2}\log\frac{2}{\delta}$, with probability at least $1 - \delta$, we have*

$$\left| \frac{1}{L}\sum_{\ell=1}^{L} rev(v^{(\ell)},x^{(\ell)},y^{(\ell)}) - \mathbb{E}_{(v,x,y)}\left[rev(v,x,y)\right] \right| \leq \epsilon.$$

*Proof.* Since $v_i \leq 1$, we have

$$rev(v, x, y) = \sum_{i=1}^{n} p_i(v, x, y) \leq \sum_{i=1}^{n} b_i(v, x, y) v_i \leq \sum_{i=1}^{n} b_i(v, x, y) \leq 1.$$

According to the Hoeffding's inequality, we have

$$\mathbb{P}\left[\left|\frac{1}{L}\sum_{\ell=1}^{L} rev(v^{(\ell)}, x^{(\ell)}, y^{(\ell)}) - \mathbb{E}_{(v,x,y)}\left[rev(v, x, y)\right]\right| \geq \epsilon\right] \leq 2\exp(-2L\epsilon^2).$$

Let $2\exp(-2L\epsilon^2) \leq \delta$, then we obtain what we need. The proof is completed. $\qquad\square$

## A.8 Orbit Averaging over Subsets of Bidders/Items

In addition, we can extended our theory to orbit averaging over the subset of the bidders/items.

**Theorem A.4.** *Let $\mathcal{Q}$ be the orbit averaging over any subset of bidders and items, and $(g, p)$ be any mechanism. Then we have*

$$\mathbb{E}_{(v,x,y)}\left[\sum_{i=1}^{n}[\mathcal{Q}p]_i(v, x, y)\right] = \mathbb{E}_{(v,x,y)}\left[\sum_{i=1}^{n} p_i(v, x, y)\right],$$

*and*

$$\mathbb{E}_{(v,x,y)}\left[\sum_{i=1}^{n} reg_i(v, x, y)\right] \geq \mathbb{E}_{(v,x,y)}\left[\sum_{i=1}^{n}[\mathcal{Q}reg]_i(v, x, y)\right],$$

*where $reg_i$ is the ex-post regret of bidder $i$.*

*Proof.* Without loss of generality, we assume that $\mathcal{Q}_1$ takes average over the first $\tilde{n}$ bidders, $\mathcal{Q}_2$ takes average over the first $\tilde{m}$ items and $\mathcal{Q}_3$ takes average over the first $\tilde{n}$ bidders and $\tilde{m}$ items. Denote $Z_1$ as $(v_{ij}, x_i : i > \tilde{n}, j \in [m])$ and $Z_2$ as $(v_{ij}, y_j : i \in [n], j > \tilde{m})$. Following Theorem 3.1, we have

$$\mathbb{E}\left[\sum_{i=1}^{n}[\mathcal{Q}_1 p]_i(v, x, y)\Big|Z_1\right] = \mathbb{E}\left[\sum_{i=1}^{n} p_i(v, x, y)\Big|Z_1\right],$$

and

$$\mathbb{E}\left[\sum_{i=1}^{n} reg_i(v, x, y)\Big|Z_1\right] \geq \mathbb{E}\left[\sum_{i=1}^{n}[\mathcal{Q}_1 reg]_i(v, x, y)\Big|Z_1\right].$$

Then, combining the fact that $\mathbb{E}[\mathbb{E}[X|Y]] = \mathbb{E}[X]$, we have

$$\mathbb{E}\left[\sum_{i=1}^{n}[\mathcal{Q}_1 p]_i(v, x, y)\right] = \mathbb{E}\left[\sum_{i=1}^{n} p_i(v, x, y)\right],$$

and

$$\mathbb{E}\left[\sum_{i=1}^{n} reg_i(v, x, y)\right] \geq \mathbb{E}\left[\sum_{i=1}^{n}[\mathcal{Q}_1 reg]_i(v, x, y)\right].$$

Similarly, replace $Z_1$ by $Z_2$, we can obtain the equations all hold for $\mathcal{Q}_2$.

Finally, we prove that $\mathcal{Q}_3 = \mathcal{Q}_1\mathcal{Q}_2 = \mathcal{Q}_2\mathcal{Q}_1$. The proof is same with the proof of Lemma 3.3. Only replace $n$ and $m$ by $\tilde{n}$ and $\tilde{m}$ respectively, and we obtain the result.

The proof is completed. $\qquad\square$

**Theorem A.5.** *Let $\mathcal{Q}$ be the orbit averaging over any subset of bidders and items, and $\mathcal{U} = \{u^{\omega} : \omega \in \Omega\}$ and $\mathcal{P} = \{p^{\omega} : \omega \in \Omega\}$ the sets of all possible utilities and payment rules. Then we have*

$$\mathcal{N}_{\infty,1}(\mathcal{Q}\mathcal{U}, r) \leq \mathcal{N}_{\infty,1}(\mathcal{U}, r) \text{ and } \mathcal{N}_{\infty,1}(\mathcal{Q}\mathcal{P}, r) \leq \mathcal{N}_{\infty,1}(\mathcal{P}, r),$$

*where $\mathcal{N}_{\infty,1}(\mathcal{U}, r)$ and $\mathcal{N}_{\infty,1}(\mathcal{P}, r)$ are the minimum numbers of balls with radius $r$ that can cover $\mathcal{U}$ and $\mathcal{P}$ under $l_{\infty,1}$-distance, respectively.*

*Proof.* Without loss of generality, we assume that $\mathcal{Q}_1$ takes average over the first $\tilde{n}$ bidders, $\mathcal{Q}_2$ takes average over the first $\tilde{m}$ items and $\mathcal{Q}_3$ takes average over the first $\tilde{n}$ bidders and $\tilde{m}$ items.

We can prove the distance between two mechanisms becomes smaller after orbit averaging, *i.e.*,

$$l_{\infty,1}(\mathcal{Q}p, \mathcal{Q}p') \leq l_{\infty,1}(p, p') \ \text{ and } \ l_{\infty,1}(\mathcal{Q}u, \mathcal{Q}u') \leq l_{\infty,1}(u, u').$$

Only replace $n$ and $m$ by $\tilde{n}$ and $\tilde{m}$ in the proof of Theorem 3.5, and we obtain the results.

The proof is completed. $\qquad\square$

## A.9 Average over Subgroups

It is worth noting that our proof only replies one assumption that the valuation joint distribution is invariant under the bidder/item permutation. Consequently, we may adopt orbit averaging over any subgroup of $S_n \times S_m$, while the benefits on revenue and ex-post regret still hold. Hence, there is a trade-off between the auction mechanism performance (revenue and ex-post regret) and the computational complexity: better performance requires more computation. In addition, the choice of the subgroup can also depend on the input feature $x$ [28], which could be more flexible.

# B  Additional Experimental Details

This section presents additional experimental details and results omitted from the main text due to space limitation.

## B.1  Additional Experimental Settings

In this section, we present detailed experimental settings.

### B.1.1  Network Architectures

We first describe the RegretNet's architecture [12]. A RegretNet consists of two parts: the allocation network $g^\omega : \mathbb{R}^{nm} \to [0,1]^{nm}$ and the payment network $p^\omega : \mathbb{R}^{nm} \to \mathbb{R}^n_{\geq 0}$, both of which are modeled as three-layer fully-connected networks with *tanh* activations. Every layer in the two networks includes 100 nodes.

For each item $j$, the payment network outputs a probability vector $(g^\omega_{1j}(b), \ldots, g^\omega_{nj}(b))^T$, where $g^\omega_{ij}(b)$ is the probability of allocating the item $j$ to the bidder $i$. To avoid allocating one item over once, a feasible allocation network needs to satisfy $\sum_{i=1}^n g^\omega_{ij}(b) \leq 1$ for all $j \in [m]$, $\omega \in \Omega$, and $b \in \mathcal{V}^{nm}$. Therefore, we compute the allocation via a *softmax activation function*. In addition, to present the probability that the item is reserved, an extra dummy node is included in the softmax computation.

To ensure the *individual rational* condition, the payment network $p^\omega$ is required to output a payment vector $p^\omega(b)$, such that $p^\omega_i(b) \leq \sum_{j=1}^m g^\omega_{ij}(b)b_{ij}$ for all $i \in [n]$. Therefore, the payment network first computes a fractional payment $\overline{p}^\omega_i(b) \in [0,1]$ for each bidder $i$ using a sigmoidal unit. Then, the final payment of the bidder $i$ is

$$p^\omega_i(b) = \overline{p}^\omega_i(b) \sum_{j=1}^m g^\omega_{ij}(b)b_{ij} \leq \sum_{j=1}^m g^\omega_{ij}(b)b_{ij}.$$

An overview of the RegretNet's architecture is illustrated in the following Figure 1.

RegretNet-PE is designed by modifying RegretNet. We adopt the allocation rule as $\widetilde{g}^\omega = \mathcal{Q}_3 g^\omega$ and the payment rule as $\widetilde{p}^\omega = \mathcal{Q}_3 p^\omega$, respectively. In this way, we may guarantee that in RegretNet-PE, the allocation is feasible and the mechanism is *individual rational*, *i.e.*, $\sum_{i=1}^n \widetilde{g}^\omega_{ij}(b) \leq 1$, and $\widetilde{p}^\omega_i(b) \leq \sum_{j=1}^m \widetilde{g}^\omega_{ij}(b)b_{ij}$. We may also show that the RegretNet-PE is always permutation-equivariant and has the same number of coefficients as the RegretNet. The proof can be found in Appendix A.1.

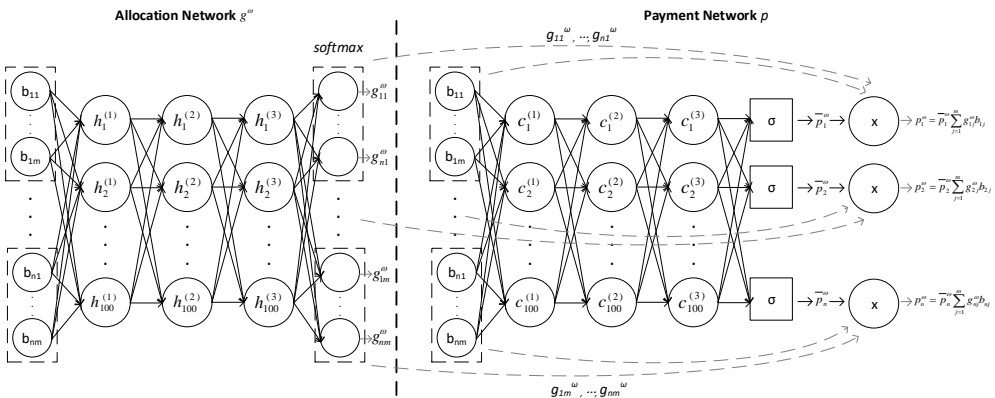

Figure 1: The allocation network $g^\omega$ and the payment network $p^\omega$ of the RegretNet with the overall mechanism parameter $\omega = (\omega_g, \omega_p)$.

### B.1.2 Training Procedures

We adopt the augmented Lagrangian method to minimize the following object function with a quadratic penalty term for violating the constraints,

$$\mathcal{L}_\rho(\omega, \lambda) = -\frac{1}{L} \sum_{l=1}^{L} \sum_{i=1}^{n} p_i^\omega(v^{(l)}) + \sum_{i=1}^{n} \lambda_i \widehat{reg}_i(\omega) + \frac{\rho}{2} \Big( \sum_{i=1}^{n} \widehat{reg}_i \Big)^2,$$

where $L$ is the number of samples, $\lambda$ is a vector of Lagrange multipliers, and $\rho > 0$ is a parameter to control the weight of the quadratic penalty. We alternately update the overall mechanism parameter $\omega$ and the Lagrange multiplier $\lambda$ as follows:

(a) $\omega^{new} \in \arg\min_\omega \mathcal{L}_\rho(\omega^{old}, \lambda^{old})$ (every iteration);

(b) $\lambda_i^{new} = \lambda_i^{old} + \rho \cdot \widehat{reg}_i(\omega^{new}), \forall i \in [n]$ (every $T_\lambda$ iterations).

The training procedure is described in the following Algorithm 1.

We divide all training samples $\mathcal{S}$ into $T$ batches $\mathcal{S}_1, \dots, \mathcal{S}_T$ of size $B$. At iteration $t$, we use the batch $\mathcal{S}_t = \{v^{(1)}, \dots, v^{(B)}\}$.

The update (a) is computed via Adam. The gradient of $\mathcal{L}_\rho$ w.r.t. $\omega$ for a fixed $\lambda^t$ is as below,

$$\nabla_\omega \mathcal{L}_\rho(\omega, \lambda^t) = -\frac{1}{B} \sum_{l=1}^{B} \sum_{i=1}^{n} \nabla_\omega p_i^\omega(v^{(l)}) + \sum_{l=1}^{B} \sum_{i=1}^{n} \lambda_i^t g_{l,i}^t + \rho \sum_{l=1}^{B} \sum_{i=1}^{n} \widehat{reg}_i(\omega) g_{l,i}^t,$$

where

$$\widehat{reg}_i(\omega) = \frac{1}{B} \sum_{l=1}^{B} \max_{v_i' \in \mathcal{V}^m} u_i^\omega(v_i^{(l)}, (v_i', v_{-i}^{(l)})) - u_i^\omega(v_i^{(l)}, v^{(l)}),$$

and

$$g_{l,i}^t = \nabla_\omega \Big[ \max_{v_i' \in \mathcal{V}^m} u_i^\omega(v_i^{(l)}, (v_i', v_{-i}^{(l)})) - u_i^\omega(v_i^{(l)}, v^{(l)}) \Big] \Big|_{\omega = \omega^t}.$$

Because $\widehat{reg}_i(\omega)$ and $g_{l,i}^t$ both contain a "max" over misreports[2], we use another Adam to compute the approximated best biddings $v'^{(l)}$. In each update on $\omega^t$, we perform $R$ updates to compute a best bidding $v_i'^{(l)}$ for each $i \in [n]$. In particular, we maintain the misreports $v_i'^{(l)}$ for each sample $l$ as the initial value in the next iteration. Then, we use these biddings $v_i'^{(l)}$ to compute the gradient $\nabla_\omega \mathcal{L}_\rho(\omega, \lambda^t)$ and then, update $\omega^t$ as $\omega^{t+1} = \omega^t - \eta \nabla_\omega \mathcal{L}_\rho(\omega^t, \lambda^t)$. After every $T_\lambda$ iterations, we update $\lambda^t$ as $\lambda_i^{t+1} = \lambda_i^t + \rho \widehat{reg}_i(\omega^{t+1})$. In addition, we increase the value of $\rho$ every a certain number of iterations, where we set the value of $\rho_t$ in each iteration $t$ prior to training.

---

[2]The misreport refers to an arbitrary bid, rather than restricted to be a truthful bid [12].

**Algorithm 1:** RegretNet and RegretNet-PE Training

**Input:** Batches $\mathcal{S}_1, \ldots, \mathcal{S}_T$ of size $B$
**Parameters:** $\forall t \in [T], \rho_t > 0, \gamma > 0, \eta > 0, T \in \mathbb{N}, R \in \mathbb{N}, T_\lambda \in \mathbb{N}$
**Initialize:** $\omega^0 \in \mathbb{R}^d, \lambda^0 \in \mathbb{R}^n$
**for** $t = 0$ **to** $T$ **do**
    Receive batch $S_t = \{v^{(1)}, \ldots, v^{(B)}\}$
    Initialize misreports $v'^{(l)}_i \in \mathcal{V}^m, \forall l \in [B], i \in [n]$
    **for** $r = 0$ **to** $R$ **do**
        $\forall l \in [B], i \in [n]$:
            $v'^{(l)}_i \leftarrow v'^{(l)}_i + \gamma \nabla_{v'_i} u_i^\omega \big(v_i^{(l)}, (v'^{(l)}_i, v^{(l)}_{-i})\big)$
    **end**
    Compute ex-post regret gradient : $\forall l \in [B], i \in [n]$:
        $g_{l,i}^t \leftarrow \nabla_\omega \Big[ u_i^\omega \big(v_i^{(l)}, (v'^{(l)}_i, v^{(l)}_{-i})\big) - u_i^\omega \big(v_i^{(l)}, v^{(l)}\big) \Big] \Big|_{\omega = \omega^t}$
    Compute Lagrangian gradient using Equation 4 and update $\omega^t$:
        $\omega^{t+1} \leftarrow \omega^t - \eta \nabla_\omega \mathcal{L}_{\rho_t}(\omega^t, \lambda^t)$
    Update Lagrange multipliers $\lambda$ once in $T_\lambda$ iterations:
    **if** *t is a multiple of $T_\lambda$* **then**
        $\lambda_i^{t+1} = \lambda_i^t + \rho_t \widehat{reg}_i(\omega^{t+1}), \forall i \in [n]$
    **else**
        $\lambda^{t+1} = \lambda^t$
    **end**
**end**

Table 3: Additional experimental results. "$n \times m$ Normal" refers that there are $n$ bidders and $m$ items, and the valuation is drawn from the truncated normal distribution $\mathcal{N}(0.3, 0.1)$ in [0,1]. The true values of the ex-post regret and the generalization error (GE) are the products of the values in the table and a factor of $10^{-5}$.

| Method | $2 \times 1$ Normal | | | $3 \times 1$ Normal | | . |
| --- | --- | --- | --- | --- | --- | --- |
| | Revenue | Regret | GE | Revenue | Regret | GE |
| Optimal | 0.304 | 0 | - | 0.391 | 0 | - |
| RegretNet | 0.275 | 97.0 | 8.50 | 0.321 | 84.0 | 45.5 |
| RegretNet-Test | 0.275 | 95.2 | - | 0.321 | 75.0 | - |
| RegretNet-PE | 0.276 | 85.4 | 8.40 | 0.382 | 69.7 | 27.6 |

| Method | $2 \times 2$ Normal | | | $5 \times 3$ Normal | | |
| --- | --- | --- | --- | --- | --- | --- |
| | Revenue | Regret | GE | Revenue | Regret | GE |
| RegretNet | 0.577 | 343 | 246 | 1.05 | 114 | 77.0 |
| RegretNet-Test | 0.577 | 327 | - | 1.05 | 32.0 | - |
| RegretNet-PE | 0.577 | 318 | 77.0 | 1.09 | 75.0 | 70.0 |

### B.1.3 Test Settings

To verify our Theorem 3.1 and Theorem 3.4, we first train a RegretNet and then project the will-trained RegretNet to be permutation-equivariant through bidder-item aggregated averaging $\mathcal{Q}_3$, denoted as "RegretNet-Test". To meet the *symmetric valuation* condition in Theorem 3.1 and Theorem 3.4, we sample a set of valuations from the distribution, which is denoted by $\mathcal{S}$, and then, induce a set of symmetric samples $\widetilde{\mathcal{S}} = \{\sigma_n v \sigma_m : \sigma_n \in S_n, v \in \mathcal{S}, \sigma_m \in S_m\}$ for test.

Table 4: Additional experimental result, where "$n \times m$ Compound" refers that there are $n$ bidders and $m$ items, and the valuations are i.i.d. sampled from the compound distributions.

| Method | $3 \times 1$ Compound | | $5 \times 1$ Compound | |
|---|---|---|---|---|
| | Revenue | Regret | Revenue | Regret |
| RegretNet | 0.516 | $< 0.001$ | 0.329 | $< 0.001$ |
| EquivariantNet | 0.498 | $< 0.001$ | 0.311 | $< 0.001$ |
| RegretNet-PE | 0.539 | $< 0.001$ | 0.356 | $< 0.001$ |

Table 5: Additional experimental result, where "$n \times m$ Uniform" refers that there are $n$ bidders and $m$ items, and the valuations are i.i.d. sampled from the uniform distribution $U[0, 1]$.

| Method | $2 \times 5$ Uniform | | | $5 \times 3$ Uniform | | |
|---|---|---|---|---|---|---|
| | Revenue | Regret | GE | Revenue | Regret | GE |
| RegretNet | 2.24 | 104 | 86.4 | 1.56 | 28.4 | 19.4 |
| RegretNet-Test | 2.24 | 74.0 | - | 1.56 | 8.60 | - |
| RegretNet-PE | 2.38 | 89.9 | 24.9 | 1.85 | 20.1 | 11.8 |

For a RegretNet-PE, there is no difference between test on $\tilde{\mathcal{S}}$ and on $\mathcal{S}$, because

$$\frac{1}{n!m!L} \sum_{l=1}^{L} \sum_{\sigma_n \in S_n} \sum_{\sigma_m \in S_m} f(\sigma_n v_i \sigma_m) = \frac{1}{n!m!L} \sum_{l=1}^{L} \sum_{\sigma_n \in S_n} \sum_{\sigma_m \in S_m} f(v_i) = \frac{1}{L} \sum_{l=1}^{L} f(v_i),$$

for any permutation-equivariant function $f$ and a RegretNet-PE is always permutation-equivariant.

To compute the best bidding $v'_i$ for each bidder $i$, we first randomly initialize $1,000$ misreports in all settings, and then, perform $2,000$ updates on each misreport via Adam with the same settings. Finally, we choose the best one (which induces a maximal utility of the bidder $i$) as the approximated best bidding $v'_i$.

### B.1.4  Implementation Details

We train the models (RegretNet and RegretNet-PE) for up to 150 epochs with a batch size of 128 ($B = 128$) and report the early-stop results for RegretNet-PE to obtain a comparable ex-post regret. The terminal iteration numbers for RegretNet-PE are $10,000$ in the $2 \times 1$ setting, $17,000$ in the $3 \times 1$ setting, $18,000$ in the $5 \times 1$ setting, $300,000$ in the $1 \times 2$ setting, and $600,000$ in the $2 \times 2$ setting. Our insight is that the larger terminal iteration number required in the $2 \times 2$ setting is because of the small model size, *i.e.*, where the networks have three layers (each of 100 nodes). The value of $\rho$ is initialized as 1.0 and increased by 5 every 200 batches. For each update on $\omega^t$, we initialize one misreport and update the misreport by Adam for each bidder with 25 steps ($R = 25$) and learning rate 0.1 ($\gamma = 0.1$). The final optimal misreports will be used to initialize the misreports for the same batch in the next epoch. We update $\omega^t$ via Adam for every batch with a learning rate of 0.001. Besides, we update $\lambda^t$ every 200 batches.

### B.2  Additional Experiment Results

We present additional experimental results. Each valuation $v_{ij}$ is sampled independently from (1) a truncated normal distribution $\mathcal{N}(0.3, 0.1)$ in $[0, 1]$; (2) a compound distribution $\mathcal{N}(\frac{x_i}{6}, 0.1)$ truncated in $[0, 1]$, where $x_i$ is sampled independently and uniformly from $\{1, 2, 3, 4, 5\}$ (cf. Setting A, [10]); and (3) a compound distribution $U[0, Sigmoid(x_i^T y_j)]$, where $x_i$ and $y_j$ are sampled independently and uniformly from $[-1, 1]$ (Setting C, [10]). All results are shown in Table 3 and Table 4. The revenue and ex-post regret of RegretNet and EquivariantNet in Table 4 come from the previous work [10]. In Table 4, we report the ex-post regret as "$< 0.001$" following the previous works.

Moreover, we extend our experiments to more complex settings, including two-bidder five-item and five-bidder three-item settings. Due to the computation limitations, we sample $\{3840, 1280\}$ data

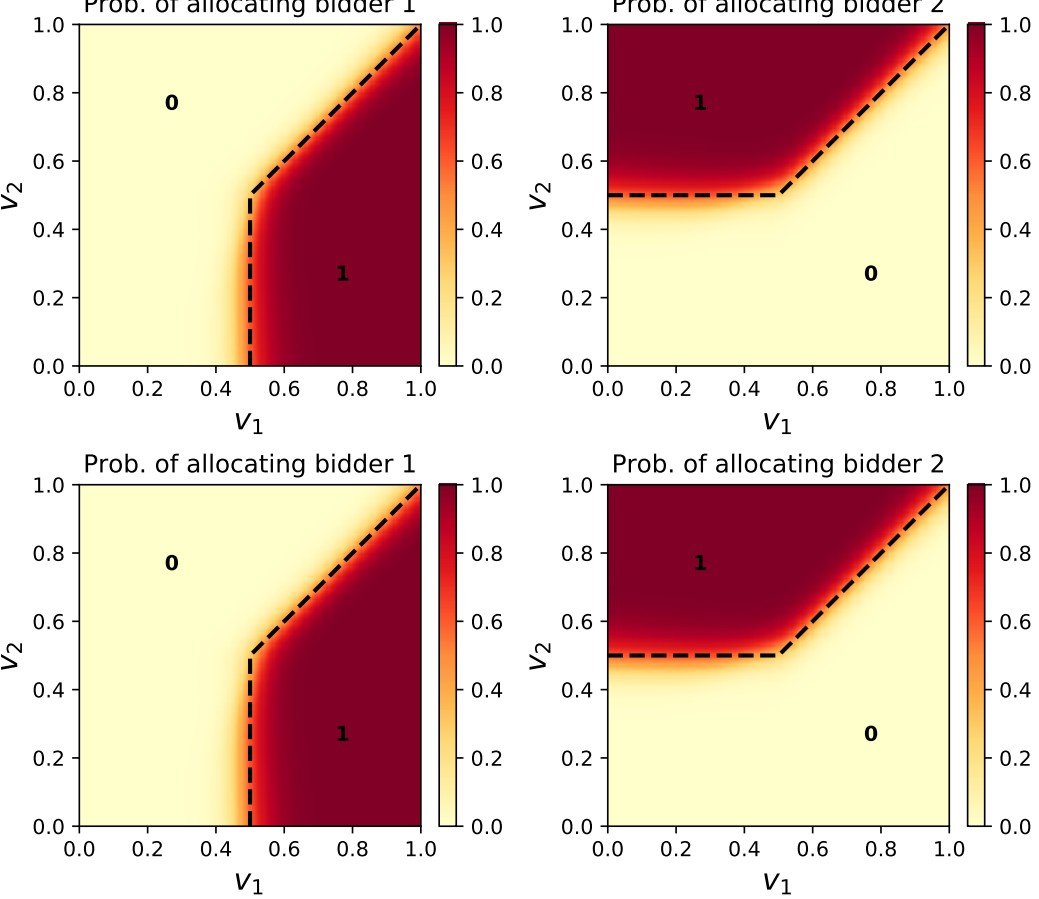

Figure 2: Allocation rule learned by RegretNet (up) and RegretNet-PE (down) for two-bidder and one-item setting. The solid regions describe the probability to allocating the item to bidder 1 (left) and bidder 2 (right). The optimal auction mechanism is described by the regions separated by the dashed black lines, where the number 0 or 1 is the probability of optimal allocation rule in the region.

points and initialize $\{150, 120\}$ misreports for test. Each valuation is sampled from the uniform distribution $U[0, 1]$ and the truncated normal distribution $\mathcal{N}(0.3, 0.1)$ in $[0, 1]$. The results are shown in Tables 3 and 5.

### B.3 Allocation Rules Learned by RegretNet and RegretNet-PE

In this section, we show the allocation rules learned by RegretNet and RegretNet-PE in two-bidder, one-item setting and one-bidder, two-item setting, where the valuation is drawn from the uniform distribution $U[0, 1]$. The optimal auction mechanisms are both known.

#### B.3.1 Two-bidder and One-item Setting

For the two-bidder, one-item setting, the optimal mechanism is well-known as Myerson auction [23], which allocates the item to the highest bidder with receiving a payment of the maximum of the second price and the reserve price, if the highest bid is higher than the reserve price. The allocation rules learned by RegretNet and RegretNet-PE are shown in Figure 2. From Figure 2. We can find that the two learned allocation rules are both almost the same as the optimal mechanism.

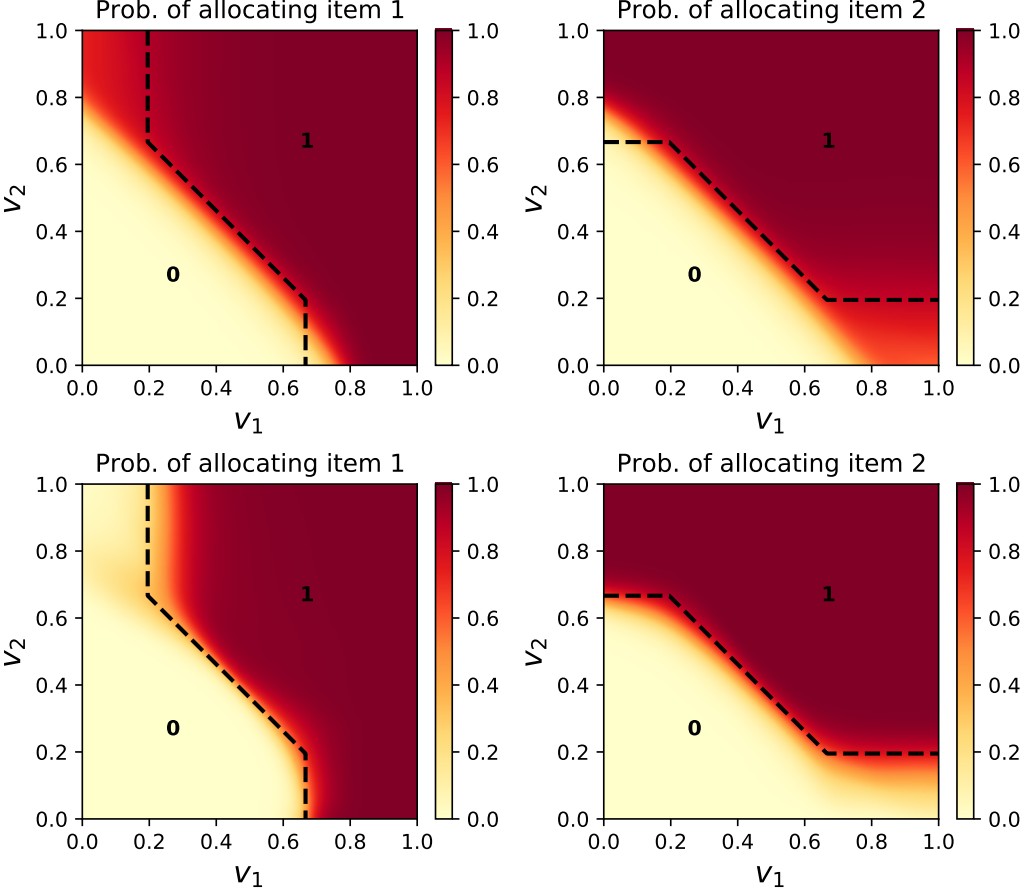

Figure 3: Allocation rule learned by RegretNet (up) and RegretNet-PE (down) for one-bidder and two-item setting. The solid regions describe the probability of allocating the first item (left) and the second item (right). The optimal auction mechanism is described by the regions separated by the dashed black lines, where the number 0 or 1 is the probability of optimal allocation rule in the region.

### B.3.2 One-bidder and Two-item Setting

The optimal mechanism is given by [20]. Same with the above, we show the allocation rules learned by RegretNet and that learned by RegretNet-PE in Figure 3. The improvement is significant. when one item's valuation is close to 0 and another item's is close to 1, the mechanism learned by RegretNet has a positive probability to allocate the item with the lower valuation to the bidder, while RegretNet-PE and the optimal mechanisms would not.