# OpenReview forum: "Benefits of Permutation-Equivariance in Auction Mechanisms"
_NeurIPS.cc/2022/Conference — NeurIPS 2022 Accept_

### Official Review · Reviewer_99t4 · 2022-07-08

**Rating:** 6
**Confidence:** 4
**Soundness:** 3 good
**Presentation:** 3 good
**Contribution:** 2 fair

**Summary:**

This work studies the benefits of incorporating permutation-equivariance into a neural network (NN) trained to function as an auction mechanism in auctions with additive valuations and symmetric bidders and/or items. The authors prove that incorporating permutation-equivariance which reflects the bidder/item symmetries 1) decreased the expected ex-post regret (measures "distance" to the space of dominant strategy incencitve compatible, "truthful", auctions) of the bidders, 2) maintains the expected auctioneer revenue, and 3) improves mechanism generalizability (e.g, to other bidder valuation profiles), intuitively via a reduction of the auction mechanism hypothesis space. Permutation-equivariance is achieved by "orbit" averaging the NN outputs (item allocations and bidder payments) over all possible permutations of bidders/items. Experiments are conducted in both single-item (theoretically optimal mechanism is known) and multi-item auctions. Properties 1, 2, and 3 are supported by the experiments in all 2, 3, and 5 bidder auctions.

**Questions:**

- In Sec 3.1.2, should there be a $\sigma_m^{-1}$ on the right side of the definition of $Q_3 p(v, x, y)$? Same with $Q_2 p(v, x, y)$ in Sec 3.1.1? If not, why not?
- In Table 1, does the, for example, "2 x 1 Uniform" header mean "number of bidders x number of items Distribution of valuations"? I did not see this explained anywhere.
- Is RegretNet-PE your model? I did not see your model named in the text anywhere (maybe I missed it) and had to infer this.
- The conclusions you draw from the experimental results appear to be over simplifications. Your first claim (1) appears to be violated in the 5 x 1 setting (your revenue is 0.633 vs optimal is 0.667). Your second claim (2) is also violated in the 5 x 3 setting (ex-post regret increases vs RegretNet, yours is 31.2 vs RegretNet 28.4). Please report accurate conclusions. If I'm misunderstanding your table, please clarify.
- How would you apply your approach to settings where subsets of the items/bidders are symmetric? Can you just average over the permutations within the subset? This might be a nice note to add to the exposition.
- Can you please compare/contrast RegretNet-PE with EquivariantNet? More generally, can you list the contributions of your work that go beyond [25]?

**Limitations:**

Can you comment on the computational expense of your approach? Averaging over all possible permutations presumably increases the number of forward passes of your NN by a factorial amount. Therefore, this approach is only feasible when the number of symmetric bidders/items is small.

**Strengths And Weaknesses:**

This work proves several properties conjectured and partially explored empirically in previous work [25]. Most of the exposition is clear. Overall, given the existence of the previous work [25] at AAAI, I find the work a bit incremental and, therefore, lacks significance beyond [25]. The authors do not compare / contrast the permutation-equivariant NN in [25] with their own (unclear if RegretNet-PE is original), and only include one comparison with that model at the end of their appendix (also without any discussion of EquivariantNet). The theorems proven in this paper are interesting, but not very surprising (esp generalizability, reducing the hypothesis space typically improves generalizability). It's unclear how useful the sample complexity bounds are and experiments are not conducted to empirically confirm the complexity bound.

---

> ### Author Response · Authors · 2022-08-02
> **Response to Reviewer 99t4 (4/4)**
>
> **Q9.1:** *Can you please compare/contrast RegretNet-PE with EquivariantNet? Can you comment on the computational expense of your approach? Averaging over all possible permutations presumably increases the number of forward passes of your NN by a factorial amount. Therefore, this approach is only feasible when the number of symmetric bidders/items is small.*
>
> **A9.1:** Thanks and addressed. We would love to emphasize that RegretNet-PE is designed to empirically validate our theory only. It has the same architecture with RegretNet, and thus allows conducting rigorous control experiments. We do not claim the contribution of novel algorithm designing, which is not the focus of this paper. For a detailed comparison; please refer to A1.
>
> **Q9.2:** *More generally, can you list the contributions of your work that go beyond [26]?*
>
> **A9.2:** For a detailed statement of our contributions; please refer to A1.
>
> [3] M.-F. F. Balcan, T. Sandholm, and E. Vitercik. Sample complexity of automated mechanism design. Neural Information Processing Systems (NeurIPS), 2016.
>
> [5] R. Cole and T. Roughgarden. The sample complexity of revenue maximization. In ACM symposium on Theory of computing, 2014.
>
> [10] Z. Duan, J. Tang, Y. Yin, Z. Feng, X. Yan, M. Zaheer, and X. Deng. A context-integrated transformer-based neural network for auction design. arXiv preprint arXiv:2201.12489, 2022.
>
> [26] J. Rahme, S. Jelassi, J. Bruna, and S. M. Weinberg. A permutation-equivariant neural network architecture for auction design. In Proceedings of the AAAI Conference on Artificial Intelligence, 2021.
>
> [28] T. Sandholm and A. Likhodedov. Automated design of revenue-maximizing combinatorial auctions. Operations Research, 2015.
>
> [29] A. Sannai, M. Imaizumi, and M. Kawano. Improved generalization bounds of group invariant/equivariant deep networks via quotient feature spaces. In International Conference on Machine Learning (ICML), 2021.
>
> [30] V. Syrgkanis. A sample complexity measure with applications to learning optimal auctions. Neural Information Processing Systems (NeurIPS), 2017.

---

> > ### Comment · Reviewer_99t4 · 2022-08-03
> > **Thank you + Follow-up Request**
> >
> > Thank you for your thorough response and for updating the manuscript. I agree with reviewers FBeS and SPL9 that it would greatly improve the paper if you:
> > 1) added a more comprehensive discussion and acknowledgement of previous work [25] (i.e., clearly set the stage that you aim to prove theoretical results about a pre-existing approach),
> > 2) explained how the EquivariantNet and RegretNet-PE models differ,
> > 3) included the results against EquivariantNet in the main body of the paper (e.g., Table 1).

---

> > > ### Author Response · Authors · 2022-08-05
> > > **Thank you! Revision completed (2/2)**
> > >
> > > **Q3:** *Included the results against EquivariantNet in the main body of the paper (e.g., Table 1).*
> > >
> > > **A3:** Thanks for your advice! We have moved the results against EquivariantNet. Please see below.
> > >
> > > In the 4-th paragraph of Section 4, p. 8, We add the following sentence and table, highlighted as red:
> > >
> > > “Also, we compare RegretNet-PE and EquivariantNet in the one-bidder, two-item setting, and the two-bidder, two-item setting.”
> > >
> > > **Table 2: Experimental results. $n \times m$ $Uniform$" refers that there are n bidders and m items, and the valuations are i.i.d. drawn from the uniform distribution $U[0,1]$.**
> > > |     Method     | $1\times 2$ $Uniform$ | $2\times 2$ $Uniform$ |
> > > | :------------: | :-------------------: | :-------------------: |
> > > |       -        |   Revenue \| Regret   |   Revenue \| Regret   |
> > > |   RegretNet    |   0.549 \| 0.00069    |   0.870 \| 0.00070    |
> > > | EquivariantNet |   0.551 \| 0.00013    |   0.873 \| 0.00100    |
> > > | RegretNet-Test |   0.549 \| 0.00062    |   0.870 \| 0.00054    |
> > > |  RegretNet-PE  |   0.643 \| 0.00022    |   0.913 \| 0.00067    |
> > >
> > > In the 9-th paragraph of Section 4, p. 9, we add the following discussion, highlighted as red:
> > >
> > > “From Tables 1 and 2, we observe that (1) compared to RegretNet, RegretNet-PE has a significantly higher revenue with a lower ex-post regret, and narrows the generalization gap between the training ex-post regret and its test counterpart; (2) compared to RegretNet, RegretNet-Test receives the same revenue with a significantly lower ex-post regret; and (3) under comparable ex-post regrets, RegretNet-PE has a considerably higher revenue than EquivariantNet, while all permutation-equivariant models (EquivariantNet, RegretNet-Test, and RegretNet-PE) can outperform RegretNet. These results show significant benefits of permutation-equivariance on revenue, ex-post regret, and generalizability, which fully supports our theoretical findings in Theorems 3,1, 3.4, and 3.5.”

---

> > > ### Author Response · Authors · 2022-08-05
> > > **Thank you! Revision completed (1/2)**
> > >
> > > Thank you very much for your reply! We have carefully improved our manuscript according to your comments. We sincerely hope your concerns will be addressed.
> > >
> > > Details are given below.
> > >
> > > **Q1:** *Added a more comprehensive discussion and acknowledgement of previous work [26], (i.e., clearly set the stage that you aim to prove theoretical results about a pre-existing approach).*
> > >
> > > **A1:** Thanks for your advice! We will carefully acknowledge and appreciate the contributions made by the previous work [26] that designs the first permutation-equivariant neural network-based auction design algorithm. We will also clarify our contributions in theoretically explaining the advantages of permutation equivariance in auction design. Please kindly see below.
> > >
> > > In the “Related works” part of Section 1, p. 2, we have added the following paragraph, highlighted as red:
> > >
> > > “Rahme et al. [26] propose the first equivariant neural network-based auction mechanism design method with significant empirical advantages. Ivanov et al. [16] propose a RegretFormer which (1) introduces attention layers to RegretNet to learn permutation-equivariant auction mechanisms, and (2) adopts a new interpretable loss function to control the revenue-regret trade-off. Duan et al. [10] extend the applicable domain to contextual settings. All these works make remarkable contributions in designing new algorithms from the empirical aspect. However, the theoretical foundations are still elusive. To our best knowledge, our paper is the first work on theoretically studying the benefits of permutation equivariance in auction design via deep learning.”
> > >
> > > In the 4-th and 5-th paragraphs of Section 1, p. 2, we have added the following paragraphs, highlighted as red:
> > >
> > > “We demonstrate that permutation-equivariant models have significant advantages in learning the optimal auction mechanism as follows. (1) We prove that the permutation-equivariance in auction mechanisms decreases the expected ex-post regret while maintaining the expected revenue invariant. Conversely, and equivalently, the permutation-equivariance promises a larger expected revenue, when the expected ex-post regret is fixed. (2) We show that the permutation-equivariance of auction mechanisms reduces the required sample complexity for a desirable generalizability. We prove that the $l_{\infty, 1}$-distance between any two mechanisms in the mechanism space decreases when they are projected to the permutation-equivariant mechanism (sub-)space. This smaller distance implies a smaller covering number of the permutation-equivariant mechanism space, which further leads to a small generalization bound [12].
> > >
> > > “We further provide an explanation for the learning process of non-permutation-equivariant neural networks (NPE-NNs). In learning the optimal auction mechanism by an NPE-NN, we show that an extra positive term exists in the quadratic penalty of the ex-post regret based on the result (1). This term serves as a regularizer to penalize the “non-permutation-equivariance”. Moreover, this regularizer also interferes the revenue maximization, and thus affects the learning performance of NPE-NNs. This further explains the advantages of permutation-equivariance in auction design.”
> > >
> > > **Q2:** *Explained how the EquivariantNet and RegretNet-PE models differ*
> > >
> > > **A2:** Thanks for your advice! We will carefully discuss the difference between EquivariantNet and RegretNet-PE. Please see below.
> > >
> > > In the 3-th paragraph of Section 4, p. 8, we have added the following paragraph, highlighted as red:
> > >
> > > “**Comparison with EquivariantNet.** RegretNet adopts two feed-forward full-connected networks to learn the allocation rule and payment rule, respectively. We denote the weight matrix in the layer $\ell$ as $W^{(\ell)}$. Both EquivariantNet and RegretNet-PE inherit the architecture of RegretNet (with some modifications), but utilize different approaches to realize the permutation equivariance. EquivariantNet applies parameter-sharing in every layer during training, to constrain $W^{(\ell)}$ to be equivariant. In contrast, RegretNet-PE employs orbit averaging to be permutation-equivariant. Specifically, RegretNet-PE adopts a weight matrix $I_{K}\otimes W^{(\ell)}(\rho^T_{g_{1}}\dots\rho^T_{g_K})^T$ in the first layer, weight matrices $I_{K}\otimes W^{(\ell)}$ in the following layers, and multiples a matrix $(\rho_{g_1}^{-1},\dots,\rho_{g_K}^{-1})$ to the output layer, where $K$ is the scale of the group $G=\\{g_1,\dots,g_K\\}$, $\rho_{g_k}$ represents the permutation operator on bidders and items, $I_K$ is an identity matrix, and $\otimes$ is the Kronecker product. It is worth noting that RegretNet-PE is only designed for verifying our theory.”

---

> > > > ### Comment · Reviewer_99t4 · 2022-08-07
> > > > **Post-Revision Response**
> > > >
> > > > Thank you very much for making those changes. The context as well as your contributions are much clearer to me now. I've updated my score.
> > > >
> > > > I see one minor typo in your revision: "full-connected" --> "fully-connected".

---

> > > > > ### Author Response · Authors · 2022-08-08
> > > > > **Thank you!**
> > > > >
> > > > > Thank you very much for your suggestions and support! The typo has been addressed.

---

> ### Author Response · Authors · 2022-08-02
> **Response to Reviewer 99t4 (3/4)**
>
> **Q7:** *The conclusions you draw from the experimental results appear to be over simplifications. Your first claim (1) appears to be violated in the 5 x 1 setting (your revenue is 0.633 vs optimal is 0.667). Your second claim (2) is also violated in the 5 x 3 setting (ex-post regret increases vs RegretNet, yours is 31.2 vs RegretNet 28.4). Please report accurate conclusions. If I'm misunderstanding your table, please clarify.*
>
> **A7:** Thanks and addressed. Following your advice, we conduct additional experiments to verify our theory, including the five-bidder and one-item, and five-bidder and three-item settings, as shown below.
>
> **Five-bidder and one-item setting:**
>
> | Method         | Revenue | Regret   | GE       |
> | :------------: | :-----: | :------: | :------: |
> | RegretNet      | 0.658   | 0.000159 | 0.000064 |
> | RegretNet-Test | 0.658   | 0.000065 |    -     |
> | RegretNet-PE   | 0.677   | 0.000132 | 0.000051 |
>
> **Five-bidder and three-item setting:**
>
> |     Method     | Revenue |  Regret  |    GE    |
> | :------------: | :-----: | :------: | :------: |
> |   RegretNet    |  1.56   | 0.000284 | 0.000194 |
> | RegretNet-Test |  1.56   | 0.000086 |    -     |
> |  RegretNet-PE  |  1.85   | 0.000201 | 0.000118 |
>
> From the results, we observe the following three arguments hold: (1) RegretNet-PE has a higher revenue and a lower ex-post regret in the multi-item settings;
> (2) RegretNet-Test receives the same revenue with a significantly lower ex-post regret; and (3) RegretNet-PE narrows the generalization gap between the training revenue/ex-post regret and their test counterparts.
> These results suggest that permutation-equivariance can (1) improve the revenue, (2) reduce the ex-post regret, and (3) narrow the generalization gap, which fully support our theory.
>
> We have also carefully investigated the cases where our claims seem violated.
>
> “Claim (1) appears to be violated in the 5 x 1 setting”: this is due to sub-optimal hyper-parameters in training, such as learning rate and the factor of the quadratic regularization term.
>
> “Claim (2) is also violated in the 5 x 3 setting”: this is due to “too early stop”.
>
> After tuning the hyper-parameters and the training time, the results in these settings are in full agreement with our theory. All the code and data will be released publicly.
>
> **Q8:** *How would you apply your approach to settings where subsets of the items/bidders are symmetric? Can you just average over the permutations within the subset? This might be a nice note to add to the exposition.*
>
> **A8:** Thanks for your helpful advice! Following you suggestion, we have extended our theory to orbit averaging over the subset of the bidders/items in rebuttal as follows.
>
> **Theorem A.4:** Let $\mathcal{Q}$ be the orbit averaging over any subset of bidders and items, and $(g,p)$ be any mechanism. Then we have
>
> $\mathbb{E}\_{(v,x,y)}\bigg[ \sum\limits\_{i=1}^n [\mathcal{Q}{p}]\_i(v,x,y)\bigg]=\mathbb{E}\_{(v,x,y)}\bigg[\sum\limits\_{i=1}^n{p}\_i(v,x,y)\bigg]$,
>
> and
>
> $\mathbb{E}\_{(v,x,y)}\bigg[\sum\limits\_{i=1}^n{reg}\_i(v,x,y)\bigg]\ge\mathbb{E}\_{(v,x,y)}\bigg[\sum\limits\_{i=1}^n[\mathcal{Q}{reg}]\_i(v,x,y)\bigg]$,
>
> where $reg_i$ is the ex-post regret of bidder $i$.
>
> **Theorem A.5:** Let $\mathcal{Q}$ be the orbit averaging over any subset of bidders and items, and $\mathcal{U}= \\{u^\omega:\omega\in\Omega\\}$ and $\mathcal{P}=\\{p^\omega:\omega\in\Omega\\}$ the sets of all possible utilities and payment rules. Then we have
>
> ${\mathcal{N}}\_{\infty,1}(\mathcal{Q}\mathcal{U},r)\le\mathcal{N}\_{\infty,1}(\mathcal{U},r)$ and $\mathcal{N}\_{\infty,1}(\mathcal{Q}{\mathcal{P}},r)\le\mathcal{N}\_{\infty,1}(\mathcal{P},r)$,
>
> where $\mathcal{N}\_{\infty,1}({\mathcal{U}},r)$ and $\mathcal{N}\_{\infty,1}({\mathcal{P}},r)$ are the minimum numbers of balls with radius $r$ that can cover $\mathcal{U}$ and $\mathcal{P}$ under $l_{\infty,1}$-distance, respectively.
>
> These would significantly scale the equivariant neural network-based auction design.

---

> ### Author Response · Authors · 2022-08-02
> **Response to Reviewer 99t4 (2/4)**
>
> **Q2:** *The theorems proven in this paper are interesting, but not very surprising (esp generalizability, reducing the hypothesis space typically improves generalizability).*
>
> **A2:** We respectfully disagree. Please note that **the hypothesis space is not reduced**. Our result is that for any set of auctions, its permutation-equivariant counterpart has a smaller covering number, despite the “number” of permutation-equivariant auctions is exactly equal with the “size” of the set of auctions. Every hypothesis $(g^\omega,p^\omega)$ in the original space has its counterpart $(\mathcal{Q}g^\omega,\mathcal{Q}p^\omega)$ in the processed space by orbit averaging. We prove that **despite the “size” of hypothesis space is fixed, the hypothesis complexity reduces.** This is usually a significant and difficult questions to compare them in the literature of equivariant neural networks, for example in [29].
>
> **Q3:** *It's unclear how useful the sample complexity bounds are and experiments are not conducted to empirically confirm the complexity bound.*
>
> **A3:** We respectfully disagree. **The study of generalizability (sample complexity bounds) is in the central place of learning theory** [3,5,29,30]. It has shown significant practical values in wide areas throughout the history of machine learning.
>
> Further, the study of generalizability has been shown a difficult question in the literature of equivariant neural networks, for example in [29]. To our best knowledge, **our sample complexity bounds is exactly the first work that endeavors to explain the benefits of permutation-equivariant auction designs from the perspective of generalizability**. The numerical study of our complexity bound is a very interesting direction but out of the scope of this paper. We will definitely study it in the future.
>
> **Q4:** *In Sec 3.1.2, should there be a $\sigma_m^{-1}$ on the right side of the definition of $\mathcal{Q}_3p(v,x,y)$? Same with $\mathcal{Q}_2p(v,x,y)$ in Sec 3.1.1? If not, why not?*
>
> **A4:** Thanks. Please note that $p$ is a $n\times 1$ matrix $(p_1,\dots,p_n)^T$, where $p_i=\sum_{j=1}^mp_{i j}$ for each $i\in[n]$. We have highlighted this in Section 3, p.4. Below is a brief proof.
>
> $
> (\mathcal{Q}\_2p)\_i(v,x,y)=\sum\limits\_{j=1}^m(\mathcal{Q}\_2p)\_{ij}(v,x,y)=\frac{1}{m!}\sum\limits\_{\sigma\_m\in S_m}\sum\limits\_{j=1}^mp\_{i\sigma\_{m}^{-1}(j)}(v\sigma\_m,x,y\sigma\_m)=
> $
>
> $\frac{1}{m!}\sum\limits\_{\sigma\_m\in S\_m}\sum\limits\_{j=1}^mp\_{i j}(v\sigma\_m,x,y\sigma\_m)=\frac{1}{m!}\sum\limits\_{\sigma\_m\in S_m}p\_i(v\sigma\_m,x,y\sigma\_m).
> $
>
> Then we have
>
> $
> \mathcal{Q}\_2p(v,x,y)=[(\mathcal{Q}\_2p)\_i(v,x,y)]\_{i=1}^n=\bigg[\frac{1}{m!}\sum\limits\_{\sigma_m\in S_m}p\_i(v\sigma_m,x,y\sigma_m)\bigg]\_{i=1}^n=\frac{1}{m!}\sum\limits\_{\sigma_m\in S_m}p(v\sigma_m,x,y\sigma_m).
> $
>
> Besides, $\mathcal{Q}\_3p(v,x,y)=\mathcal{Q}\_1(\mathcal{Q}\_2p)(v,x,y)=\mathcal{Q}\_1\Big[\frac{1}{m!}\sum\limits\_{\sigma_m\in S_m}p(v\sigma_m,x,y\sigma_m)\Big]=\frac{1}{n!m!}\sum\limits\_{\sigma_n\in S\_n}\sum\limits\_{\sigma_m\in S\_m}\sigma\_n^{-1}p(\sigma\_n v\sigma_m,\sigma\_n x,y\sigma_m).$
>
> The proof is completed.
>
> **Q5:** *In Table 1, does the, for example, "2 x 1 Uniform" header mean "number of bidders x number of items Distribution of valuations"? I did not see this explained anywhere.*
>
> **A5:** Thanks and addressed. Yes, you are right. We have added a detailed explanation in the title of Table 1.
>
> **Q6:** *Is RegretNet-PE your model? I did not see your model named in the text anywhere (maybe I missed it) and had to infer this.*
>
> **A6:** Yes, it is. We respectfully note that an explanation has been included in the original paper; please refer to Section 1, p. 2:
>
>  “We design permutation equivariant versions of RegretNet (RegretNet-test and RegretNet-PE) by projecting the RegretNet to the permutation-equivariant mechanism space in the training and test stage.”
>
> and Appendix B.1.1, p. 27:
>
> “RegretNet-PE is designed by modifying RegretNet. We adopt the allocation rule as $\widetilde{g}^\omega=\mathcal{Q}_3g^\omega$ and the payment rule as $\widetilde{p}^\omega=\mathcal{Q}_3p^\omega$, respectively.”.
>
> To avoid any confusion, we have added a further explanation in Section 4, p.8:
>
> “We project RegretNet to the permutation-equivariant mechanism space via orbit averaging: (1) using bidder averaging for the bidder-symmetry condition, (2) applying item averaging for the item-symmetry condition, and (3) employing bidder-item aggregated averaging for the bidder-symmetry and item-symmetry condition. The projected model is called RegretNet-PE.”

---

> ### Author Response · Authors · 2022-08-02
> **Response to Reviewer 99t4 (1/4)**
>
> Thank you for your thorough review and constructive comments. All your concerns have been carefully responded below. The manuscript is carefully revised accordingly. We sincerely hope our responses fully address your questions.
>
>
>
> **Q1:** *Overall, given the existence of the previous work [26] at AAAI, I find the work a bit incremental and, therefore, lacks significance beyond [26]. The authors do not compare / contrast the permutation-equivariant NN in [26] with their own (unclear if RegretNet-PE is original), and only include one comparison with that model at the end of their appendix (also without any discussion of EquivariantNet).*
>
> **A1:** We respectfully disagree. We would love to clarify our contributions as follows:
>
> * **This is the first work that theoretically validates the advantages of equivariance in auction design, which has been widely recognized as an timely and important problem with both academic and practical significance**, see, _e.g._, [10, 12, 26].
> We prove that
>
>  * **Permutation equivariance decreases the expected ex-post regret and increases the expected revenue** (Th. 3.1). This theorem inspires to measure the distance between a mechanism and the optimal permutation-equivariant mechanism by the difference of their ex-post regrets when the revenue is fixed. This may have values to inspire novel algorithms for learning the target permutation-equivariant mechanism.
>
>  * **Permutation equivariance improves the generalizability** (see Th. 3.5 and Th. 3.6). This suggests that permutation-equivariant neural networks-based methods have better decision-making performance in unseen scenarios, when the training performance and the sample size are fixed. This helps design sample-efficient algorithms.
>
>  **Our theory explain previously reported experimental results** [10, 26]. Th. 3.1 explains “both CITransNet and CIEquivariantNet outperform CIRegretNet a lot in all the 3×10 and 5×10 auctions” [10]. Th. 3.5 and Th. 3.6 explain the phenomenon reported in [26] that “when fewer training samples are available, our equivariant architecture (EquivariantNet) generalizes while RegretNet struggles to.”
>
> * **We further provide an explanation for the learning process of non-permutation-equivariant neural
> networks** (NPE-NNs) (see Section 3.1.3). According to Th.3.1, every mechanism would not have a better performance than the projected permutation-equivariant mechanism via orbit averaging. Thus, intuitively, the permutation-equivariant mechanisms are the “optimal targets". In learning the optimal auction mechanism by an NPE-NN, we show that an extra positive term exists in the quadratic penalty of the ex-post regret based on the result (1). This term serves as a regularizer to penalize the “non-permutation-equivariance”. Moreover, this regularizer also interferes the revenue maximization, and thus affects the learning performance of NPE-NNs. This further explains the advantages of permutation-equivariance in auction design.
>
> There are only two related papers in the literature [10, 26]. Rahme et al. [26] proposes the first equivariant neural network-based auction mechanism design method with significant empirical advantages.  Duan et al. [10] extends the applicable domain to contextual settings. However, both works are only conducted from the empirical perspective.
> **We appreciate the contributions by [10, 26] on proposing permutation-equivariant auction design algorithms, but this would not undermine our contributions.**
>
> Besides, we conduct comparison experiments with [26]. The experimental results are shown as below.
>
> **One bidder and two items.** The valuations are i.i.d. drawn from the uniform distribution $U[0,1]$. The optimal mechanism is proved by [19].
>
> | Method|Revenue|Regret|Optimal|
> | :------------: | :-----: | :-----: | :-----: |
> | EquivariantNet | 0.5510  | 0.00013 | 0.5500  |
> |   ReregtNet    | 0.5491  | 0.00069 | 0.5500  |
> | RegretNet-Test | 0.5491  | 0.00062 | 0.5500  |
> |  RegretNet-PE| 0.5506  | 0.00040 | 0.5500  |
>
> **Two bidders and two items.** Each bidder is i.i.d. drawn the valuation for each item from the uniform distribution $U[0,1]$. We compare the optimal auction in $\text{VVCA}$ and $\text{AMA}_\text{bsym}$ families [28], as shown in the following table.
>
> |Method|Revenue|Regret|Optimal|
> | :------------: | :-----: | :-----: | :-----: |
> | EquivariantNet |  0.873  | 0.00100 |0.860|
> |RegretNet|0.831|0.00174|0.860|
> |RegretNet-Test|0.831| 0.00144 |0.860|
> |RegretNet-PE|0.874| 0.00099 |0.860|
>
> From the tables, we observe that (1) RegretNet-PE records the optimal revenues with negligible ex-post regrets; (2) compared to RegretNet, RegretNet-Test has a very close revenue and a smaller ex-post regret; (3) RegretNet-PE has a higher revenue and a lower ex-post regret than RegretNet; and (4) RegretNet-PE has almost the same revenues and ex-post regrets with EquivariantNet, while both of them achieve the optimal revenues with negligible ex-post regret.
>
> The results fully support our theory.

---

### Official Review · Reviewer_SPL9 · 2022-07-08

**Rating:** 5
**Confidence:** 2
**Soundness:** 3 good
**Presentation:** 1 poor
**Contribution:** 3 good

**Summary:**

This paper studied permutation-equivariant auction mechanisms, which, roughly, are auctions that do not depend on the labeling of the bidders/items. Permutation equivariant auctions are a promising solution concept for settings where the bidder/item distributions are symmetric. The authors prove that permutation equivariant auctions never decrease revenue, and always yield better ex-post regret (bringing the auction closer to being perfectly incentive compatible). The authors explore generalization guarantees, and provide experimental evidence for the benefits of permutation-equivariant auctions.

**Questions:**

How do the results of this paper compare to Rahme et al. [25]? My understanding is that the approach in this paper involves a conversion of any auction mechanism to a permutation-equivaraint one via orbit averaging. Is this a subset of the set of permutation equivariant mechanisms considered by Rahme et al?

In Theorem 3.5, wouldn’t covering number inequalities follow from the simple fact that the set of all permutation equivariant auctions is a subset of the set of all auctions? (I think this is what lines 261-263 is saying.) Is there a more quantitative gap that can be obtained?


**Strengths And Weaknesses:**

This paper studies a relevant, important, and difficult question: the design of optimal auctions. However, I think there are many points that need to be addressed. First, I found the paper to generally be difficult to read. Permutation equivariance is never given a clear definition in the auction context – instead it is abstractly defined for arbitrary maps via group representations (I don’t know if the authors should assume that the NeurIPS audience is intimately familiar with group theory jargon). Then, my understanding was that permutation equivariance is obtained by projecting any map via an orbit-averaging operator (but I wasn’t sure why this is the case). *I think that lines 157-158 more or less make this clear: the allocation function and payment are being averaged over all permutations of the bidders (or items) to obtain a symmetric auction.

This brings me to the second main concern that there is almost no comparison to previous work on symmetric auctions/study of permutation equivariance. Permutation equivariance is the main contribution of Rahme et al. [25] in the context of auction design, but there is nearly zero discussion or comparison to their results.

My reading is that the main contribution of this paper is a novel way to view permutation equivariance: any auction can be converted into a permuation equivariant one via the orbit averaging operator. If this is indeed the main contribution, I think the intuition for what is going on ought to be much more clearly spelled out (because it seems like an elegant and novel contribution!).

The overall writing style (flow/sentence structure/grammar/spelling) could also use a lot of improvement which would make the paper easier to read and follow.

---

> ### Author Response · Authors · 2022-08-02
> **Response to Reviewer SPL9 (3/3)**
>
> [10] Z. Duan, J. Tang, Y. Yin, Z. Feng, X. Yan, M. Zaheer, and X. Deng. A context-integrated transformer-based neural network for auction design. arXiv preprint arXiv:2201.12489, 2022.
>
> [19] A. M. Manelli and D. R. Vincent. Bundling as an optimal selling mechanism for a multiple-good monopolist. Journal of Economic Theory, 2006.
>
> [26] J. Rahme, S. Jelassi, J. Bruna, and S. M. Weinberg. A permutation-equivariant neural network architecture for auction design. In Proceedings of the AAAI Conference on Artificial Intelligence, 2021.
>
> [28] T. Sandholm and A. Likhodedov. Automated design of revenue-maximizing combinatorial auctions. Operations Research, 2015.
>
> [29] A. Sannai, M. Imaizumi, and M. Kawano. Improved generalization bounds of group invariant/equivariant deep networks via quotient feature spaces. In International Conference on Machine Learning (ICML), 2021.

---

> ### Author Response · Authors · 2022-08-02
> **Response to Reviewer SPL9 (2/3)**
>
> **Q3:** *My reading is that the main contribution of this paper is a novel way to view permutation equivariance: any auction can be converted into a permuation equivariant one via the orbit averaging operator. If this is indeed the main contribution, I think the intuition for what is going on ought to be much more clearly spelled out (because it seems like an elegant and novel contribution!).*
>
> **A3:** Thanks. We would love to clarify our contributions as follows:
>
> * **This is the first work that theoretically validates the advantages of equivariance in auction design**, which has been widely recognized as a timely and important problem with both academic and practical significance, see, _e.g._, [10, 12, 26].  We prove that
>
>   * **Permutation equivariance decreases the expected ex-post regret and increases the expected revenue** (Th. 3.1). This theorem inspires to measure the distance between a mechanism and the optimal permutation-equivariant mechanism by the difference of their ex-post regrets when the revenue is fixed. This may have value in inspiring novel algorithms for learning the target permutation-equivariant mechanism.
>
>   * **Permutation equivariance improves the generalizability** (see Th. 3.5 and Th. 3.6). This suggests that permutation-equivariant neural networks-based methods have better decision-making performance in unseen scenarios, when the training performance and the sample size are fixed. This helps design sample-efficient algorithms.
>
>   **Our theory explains previously reported experimental results** [10, 26]. Th. 3.1 explains “both CITransNet and CIEquivariantNet outperform CIRegretNet a lot in all the 3×10 and 5×10 auctions” [10]. Th. 3.5 and Th. 3.6 explain the phenomenon reported in [26] that “when fewer training samples are available, our equivariant architecture (EquivariantNet) generalizes while RegretNet struggles to.”
>
> * **We further provide an explanation for the learning process of non-permutation-equivariant neural
> networks** (NPE-NNs) (see Section 3.1.3). According to Th.3.1, every mechanism would not have a better performance than the projected permutation-equivariant mechanism via orbit averaging. Thus, intuitively, the permutation-equivariant mechanisms are the “optimal targets". In learning the optimal auction mechanism by an NPE-NN, we show that an extra positive term exists in the quadratic penalty of the ex-post regret based on the result (1). This term serves as a regularizer to penalize the “non-permutation-equivariance”. Moreover, this regularizer also interferes the revenue maximization, and thus affects the learning performance of NPE-NNs. This further explains the advantages of permutation-equivariance in auction design.
>
> **Q4:** *The overall writing style (flow/sentence structure/grammar/spelling) could also use a lot of improvement which would make the paper easier to read and follow.*
>
> **A4:** Thanks. We will carefully revise our manuscript following your suggestions.
>
> **Q5:** *How do the results of this paper compare to Rahme et al. [26]? My understanding is that the approach in this paper involves a conversion of any auction mechanism to a permutation-equivaraint one via orbit averaging. Is this a subset of the set of permutation equivariant mechanisms considered by Rahme et al?*
>
> **A5:** Please kindly refer to A2 and A3.
>
> **Q6:** *In Theorem 3.5, wouldn’t covering number inequalities follow from the simple fact that the set of all permutation equivariant auctions is a subset of the set of all auctions? (I think this is what lines 261-263 is saying.) Is there a more quantitative gap that can be obtained?*
>
> **A6:** We respectfully disagree. Our result is that **for any set of auctions, its permutation-equivariant counterpart has a smaller covering number, despite the “number” of permutation-equivariant auctions being exactly equal to the “size” of the set of auctions.** Every hypothesis $(g^\omega,p^\omega)$ in the original space has its counterpart $(\mathcal{Q}g^\omega,\mathcal{Q}p^\omega)$ in the processed space by orbit averaging. We prove that **despite the “size” of hypothesis space is fixed, the hypothesis complexity reduces**. This is usually a significant and difficult question to compare them in the literature on equivariant neural networks, for example in [29]. This result is strictly stronger than what you mentioned.

---

> ### Author Response · Authors · 2022-08-02
> **Response to Reviewer SPL9 (1/3)**
>
> Thank you for your thorough review and constructive comments. All your concerns have been carefully responded below. The manuscript is carefully improved accordingly. We sincerely hope our responses fully address your questions.
>
> **Q1.1:** *I found the paper to generally be difficult to read. Permutation equivariance is never given a clear definition in the auction context – instead it is abstractly defined for arbitrary maps via group representations (I don’t know if the authors should assume that the NeurIPS audience is intimately familiar with group theory jargon).*
>
> **A1.1:** Thanks. We will carefully revise our manuscript following your suggestions. Specifically, we have added a formal definition in Section 2, p. 3, as follows.
>
> **Definition 2.2 (permutaion-equivariant mapping):** A permutation-equivariant mapping is defined to be $f:\mathbb{R}^{n\times m}\to \mathbb{R}^{n\times m}$ that for any instance $x\in\mathbb{R}^{n\times m}$, and permutation matrices $\sigma_n\in\mathbb{R}^{n\times n}$ and $\sigma_m\in\mathbb{R}^{m\times m}$, we have
>
> $f(\sigma_nx\sigma_m)=\sigma_nf(x)\sigma_m$.
>
> **Q1.2:** _Then, my understanding was that permutation equivariance is obtained by projecting any map via an orbit-averaging operator (but I wasn’t sure why this is the case)._
>
> **A1.2:** Yes, but this would not lose any generality of our theory. **Every equivariant neural network can be seen as the projected network by the orbit averaging projection operator from a neural network.** Based on this, we propose an aggregation framework that averages the allocation function and payment over all permutations of the bidders (or items) to obtain a symmetric auction. This framework covers all permutation-equivariant auction mechanisms and brings significant convenience to our theoretical study.
>
> To clear any confusion, we will give the following proposition in Section 2, p. 4, as follows,
>
> **Proposition 2.3:** Orbit averaging $\mathcal{Q}$ is a projection to the equivariant mapping space $\{f:\psi\circ f=f\circ \rho\}$, *i.e.*, $\psi\circ\mathcal{Q}f=\mathcal{Q}f\circ\rho$ and $\mathcal{Q}^2=\mathcal{Q}$. In particular, if $f$ is already equivariant, then $\mathcal{Q}f=f$.
>
> A detailed proof is given in Appendix A.1.
>
> **Q2:** *This brings me to the second main concern that there is almost no comparison to previous work on symmetric auctions/study of permutation equivariance. Permutation equivariance is the main contribution of Rahme et al. [26] in the context of auction design, but there is nearly zero discussion or comparison of their results.*
>
> **A2:** Thanks and addressed. We have added a detailed and comprehensive discussion and comparison.
>
> To our best knowledge, **our paper is the first paper on theoretically explaining the benefits of permutation equivariance in auction design via deep learning.** Rahme et al. [26] proposes the first equivariant neural network-based auction mechanism design method with significant empirical advantages.  Duan et al. [10] extends the applicable domain to contextual settings. However, both works are only conducted from the empirical perspective.
> We appreciate the contributions by [10, 26] on proposing permutation-equivariant auction design algorithms, but this would not undermine our contributions. Please kindly refer to the following A3 for a detailed statement on our contributions.
>
> Besides, we conduct comparison experiments with [26]. The experimental results are shown as below.
>
> **One bidder and two items.** The valuations are i.i.d. drawn from the uniform distribution $U[0,1]$. The optimal mechanism is proved by [19].
>
> |     Method     | Revenue | Regret  | Optimal |
> | :------------: | :-----: | :-----: | :-----: |
> | EquivariantNet | 0.5510  | 0.00013 | 0.5500  |
> |   ReregtNet    | 0.5491  | 0.00069 | 0.5500  |
> | RegretNet-Test | 0.5491  | 0.00062 | 0.5500  |
> |  RegretNet-PE  | 0.5506  | 0.00040 | 0.5500  |
>
> **Two bidders and two items.** Each bidder is i.i.d. drawn the valuation for each item from the uniform distribution $U[0,1]$. We compare the optimal auction in $\text{VVCA}$ and $\text{AMA}_\text{bsym}$ families [28], as shown in the following table.
>
> |     Method     | Revenue | Regret  | Optimal |
> | :------------: | :-----: | :-----: | :-----: |
> | EquivariantNet |  0.873  | 0.00100 |  0.860  |
> |   RegretNet    |  0.831  | 0.00174 |  0.860  |
> | RegretNet-Test |  0.831  | 0.00144 |  0.860  |
> |  RegretNet-PE  |  0.874  | 0.00099 |  0.860  |
>
> From the tables, we observe that (1) RegretNet-PE records the optimal revenues with negligible ex-post regrets; (2) compared to RegretNet, RegretNet-Test has a very close revenue and a smaller ex-post regret; (3) RegretNet-PE has a higher revenue and a lower ex-post regret than RegretNet; and (4) RegretNet-PE has almost the same revenues and ex-post regrets with EquivariantNet, while both of them achieve the optimal revenues with negligible ex-post regret.
>
> The results fully support our theory.

---

> ### Author Response · Authors · 2022-08-07
> **Sincerely looking forward to your reply**
>
> Dear Reviewer SPL9,
>
> Thank you very much for your thorough comments and helpful comments. We are looking forward to your reply. If you have further concerns or requests, please kindly give us a chance to address them in the author-reviewer discussion period. If all your concerns have been resolved, it is much appreciated if you may raise the rating of our work.
>
> Thank you!
>
> The authors

---

> ### Comment · Reviewer_SPL9 · 2022-08-07
> **Response to authors**
>
> Thank you for the very detailed response, which highlights the fact that the main contribution of this paper is a theoretical justification for permutation equivariance in auction design. Thank you also for clarifying my misunderstanding re the covering number result. Given the clarification, I am raising my score to 5. However, I do think the manuscript needs a round of very careful proofreading and revising to make it readable before publication.

---

> > ### Author Response · Authors · 2022-08-07
> > **Thanks for your support and suggestions!**
> >
> > Thank you very much for your support and suggestions! We will carefully revise and proofread our paper according to your suggestions to ensure the readability.

---

### Official Review · Reviewer_FBeS · 2022-07-09

**Rating:** 7
**Confidence:** 4
**Soundness:** 4 excellent
**Presentation:** 3 good
**Contribution:** 4 excellent

**Summary:**

The use of deep neural networks for automated mechanism design is increasingly important. Several recent papers have observed that when valuation distributions are symmetric, a symmetric auction is optimal. Thus, neural networks for mechanism design ought to respect this symmetry by using architectures (i.e. DeepSets, attention) that are permutation-equivariant. The current paper theoretically analyzes the effect of equivariance: they give a technique for projecting non-equivariant mechanisms onto the set of equivariant mechanisms, and show that it has several nice properties. Most importantly, it always preserves auction revenue while weakly improving expected violations of strategyproofness. From a learning theory perspective, learning over the class of the projected mechanisms also has better generalization guarantees. And, the projection technique is constructive so the authors can actually train these mechanisms and show they perform decently.

**Questions:**

Do you have any thoughts or speculation on how orbit averaging interacts with the gradient-based computation of best untruthful bids? Is it possible that it is somehow "harder to optimize" the misreports when orbit averaging, compared to normal RegretNet?

You consider both bidder-permutation and item-permutation. In addition to comparing against the Myerson auction in the 1 item setting and looking at bidder-permutation, it might make sense to compare to some of the known multi-item mechanisms in the 1 bidder setting (e.g. the Manelli-Vincent mixed bundling mechanism as in the original RegretNet paper), and look at item-permutation. Training a 2-item 1 agent orbit averaged mechanism could be interesting in this case (where the optimal mechanism is in fact item-equivariant).

**Limitations:**

In addition to focusing a little more on previous work about equivariant auctions in the background sections, it would enhance the experiment section to include results from other equivariant auction networks where the comparison makes sense. I don't think running lots of new experiments is necessary, but where the numbers can be compared, I think this should be done.

I think the authors should further spell out how orbit averaging interacts with the allocation constraints (items sum to one). This was a question I had as I was reading the paper. I checked and saw that orbit averaging should preserve feasibility of allocations, but I think it makes sense to explicitly reassure the reader that this holds, rather than leaving the reader to check it.

**Strengths And Weaknesses:**

*Originality*

I'm not convinced this is hugely original. It seems mainly to borrow other techniques related to equivariant ML. I think that's fine -- it's putting pieces together in an interesting and important way, but there is no really new technique here.

The paper claims in one sentence to be the first work explaining the benefits of permutation equivariance in auction design via deep learning. This is arguably not true depending on how one parses the word "explaining". Previous works looking at equivariant networks for auction design are at least pointed out and cited, but I think they should be given more discussion and the claim about being "the first" should be weakened or qualified. Doing this would not detract from the significance of the paper, in fact it would enhance the motivation for the results in this paper.

*Quality*

The quality of the paper seems high, the theoretical results are reasonable, and the empirical results are convincing enough given they are not the main focus.

*Clarity*

The paper is clear. There are good proof sketches in the main text and the full proofs are easy to follow. Some of the early definitions may be too compressed and hard to follow, but I think this is the fault of the conference page limits and not the fault of the authors.

*Significance*

This is a thoughtful contribution to an important topic in auction design. There are already several papers looking at symmetry in automated mechanism design, but this is a new, clear, and useful perspective.

---

> ### Author Response · Authors · 2022-08-02
> **Response to Reviewer FBeS (2/2)**
>
> **Q4:** *In addition to focusing a little more on previous work about equivariant auctions in the background sections, it would enhance the experiment section to include results from other equivariant auction networks where the comparison makes sense. I don't think running lots of new experiments is necessary, but where the numbers can be compared, I think this should be done.*
>
> **A4:** Thanks and addressed. To our best knowledge, there are only two equivariant neural networks-based methods in the literature [10, 26]. We have conducted additional experiments to compare with [26], as shown below.
>
> **One bidder and two items.** The valuations are i.i.d. drawn from the uniform distribution $U[0,1]$. The optimal mechanism is proved by [19].
>
> |     Method     | Revenue | Regret  | Optimal |
> | :------------: | :-----: | :-----: | :-----: |
> | EquivariantNet | 0.5510  | 0.00013 | 0.5500  |
> |   ReregtNet    | 0.5491  | 0.00069 | 0.5500  |
> | RegretNet-Test | 0.5491  | 0.00062 | 0.5500  |
> |  RegretNet-PE  | 0.5506  | 0.00040 | 0.5500  |
>
> **Two bidders and two items.** Each bidder is i.i.d. drawn the valuation for each item from the uniform distribution $U[0,1]$. We compare the optimal auction in $\text{VVCA}$ and $\text{AMA}_\text{bsym}$ families [28], as shown in the following table.
>
> |     Method     | Revenue | Regret  | Optimal |
> | :------------: | :-----: | :-----: | :-----: |
> | EquivariantNet |  0.873  | 0.00100 |  0.860  |
> |   RegretNet    |  0.831  | 0.00174 |  0.860  |
> | RegretNet-Test |  0.831  | 0.00144 |  0.860  |
> |  RegretNet-PE  |  0.874  | 0.00099 |  0.860  |
>
> From the tables, we observe that (1) RegretNet-PE records the optimal revenues with negligible ex-post regrets; (2) compared to RegretNet, RegretNet-Test has a very close revenue and a smaller ex-post regret; (3) RegretNet-PE has a higher revenue and a lower ex-post regret than RegretNet; and (4) RegretNet-PE has almost the same revenues and ex-post regrets with EquivariantNet, while both of them achieve the optimal revenues with negligible ex-post regret.
>
> The results fully support our theory.
>
> **Q5:** *I think the authors should further spell out how orbit averaging interacts with the allocation constraints (items sum to one). This was a question I had as I was reading the paper. I checked and saw that orbit averaging should preserve feasibility of allocations, but I think it makes sense to explicitly reassure the reader that this holds, rather than leaving the reader to check it.*
>
> **A5:** Thanks and addressed. Yes, that orbit averaging preserves feasibility of allocations. We have added a clear statement to explicitly reassure the reader that this holds; please refer to Section 2, p. 4. A detailed proof can be found in Appendix A.1.
>
> [10] Z. Duan, J. Tang, Y. Yin, Z. Feng, X. Yan, M. Zaheer, and X. Deng. A context-integrated transformer-based neural network for auction design. arXiv preprint arXiv:2201.12489, 2022.
>
> [19] A. M. Manelli and D. R. Vincent. Bundling as an optimal selling mechanism for a multiple-good monopolist. Journal of Economic Theory, 2006.
>
> [26] J. Rahme, S. Jelassi, J. Bruna, and S. M. Weinberg. A permutation-equivariant neural network architecture for auction design. In Proceedings of the AAAI Conference on Artificial Intelligence, 2021.
>
> [28] T. Sandholm and A. Likhodedov. Automated design of revenue-maximizing combinatorial auctions. Operations Research, 2015.

---

> > ### Comment · Reviewer_FBeS · 2022-08-05
> > **Thank you**
> >
> > Thank you for your careful and detailed response, I appreciated reading it and it cleared up a lot of things.
> >
> > I also want to bring to your attention the following work: https://arxiv.org/abs/2202.13110  which is another transformer-based auction architecture (that also claims benefits from equivariance). I want to be clear I'm not asking for any extra experiments, but this probably also merits citation when discussing equivariant neural auctions in the text.

---

> > > ### Author Response · Authors · 2022-08-06
> > > **Thank you! Discussion and acknowledgement added**
> > >
> > > Thank you very much for your recognition and support! We will carefully acknowledge and appreciate its contributions in our paper. Please see the “Related works” part in Introduction, p. 2, highlighted as red:
> > >
> > > "Rahme et al. [26] propose the first equivariant neural network-based auction mechanism design method with significant empirical advantages. Ivanov et al. [16] propose a RegretFormer which (1) introduces attention layers to RegretNet to learn permutation-equivariant auction mechanisms, and (2) adopts a new interpretable loss function to control the revenue-regret trade-off. Duan et al. [10] extend the applicable domain to contextual settings. All these works make remarkable contributions in designing new algorithms from the empirical aspect. However, the theoretical foundations are still elusive. To our best knowledge, our paper is the first work on theoretically studying the benefits of permutation equivariance in auction design via deep learning."

---

> ### Author Response · Authors · 2022-08-02
> **Response to Reviewer FBeS (1/2)**
>
> Thank you very much for your constructive comments and kind support! All your concerns have been carefully responded below. The manuscript is carefully revised accordingly. We sincerely hope our responses fully address your questions.
>
> **Q1:** *The paper claims in one sentence to be the first work explaining the benefits of permutation equivariance in auction design via deep learning. This is arguably not true depending on how one parses the word "explaining". Previous works looking at equivariant networks for auction design are at least pointed out and cited, but I think they should be given more discussion and the claim about being "the first" should be weakened or qualified.*
>
> **A1:** Thanks and addressed. We have toned down the statement to “this is the first work on *theoretically* explaining the benefits of permutation equivariance in auction design via deep learning.”
>
> To our best knowledge, only [10,26] have studied the benefits of permutation equivariance in automated auction design via deep learning; however, all of them are from the empirical perspectives.
>
> **Q2:** Do you have any thoughts or speculation on how orbit averaging interacts with the gradient-based computation of best untruthful bids? Is it possible that it is somehow "harder to optimize" the misreports when orbit averaging, compared to normal RegretNet?
>
> **A2:** Thanks. In many practical cases, it is “easier” to optimize the misreports in the orbit averaging settings; the required computational cost is reduced. In some others, the required computational cost increases when orbit averaging is applied.
> Whether it is "harder to optimize" in the case of orbit averaging depends on the original objective functions. It is an interesting direction to study the exact conditions of whether the required computational cost would increase.
> Examples are shown below.
>
> **The required computational cost decreases:**
>
> Suppose the original objective function (the utility function for each bidder) is $f(x,y)=y\*x^2-y^2\*x+2\*x$, where we assume $x$ and $y$ are the misreport of the first bidder and the item’s valuation for the second bidder, respectively. It becomes $g(x,y)=(f(x,y)+f(y,x))/2=x+y$ after orbit averaging. The computational cost is reduced.
>
> **The required computational cost increases:**
>
> Suppose the objective function is $f(x,y)=2y^2\*x$. It becomes $g(x,y)=(f(x,y)+f(y,x))/2=y\*x^2+y^2\*x$ after orbit averaging. The computational cost increases.
>
> **Q3:** *In addition to comparing against the Myerson auction in the 1 item setting and looking at bidder-permutation, it might make sense to compare to some of the known multi-item mechanisms in the 1 bidder setting (e.g. the Manelli-Vincent mixed bundling mechanism as in the original RegretNet paper), and look at item-permutation. Training a 2-item 1-agent orbit averaged mechanism could be interesting in this case (where the optimal mechanism is in fact item-equivariant).*
>
> **A3:** Thanks for your advice. We have conducted additional experiments on the 1-agent 2-item settings following your suggestions. The item valuations are i.i.d. drawn from the uniform distribution $U[0,1]$. The experimental results are shown below.
>
> |     Method     | Revenue | Regret  | Optimal |
> | :------------: | :-----: | :-----: | :-----: |
> |   RegretNet    | 0.5491  | 0.00069 | 0.5500  |
> | RegretNet-Test | 0.5491  | 0.00062 | 0.5500  |
> |  RegretNet-PE  | 0.5506  | 0.00040 | 0.5500  |
>
> From the table, we observe that: (1) RegretNet-Test has the same revenue and a smaller ex-post regret with RegretNet; and (2) RegretNet-PE has a higher revenue and a lower ex-post regret compared with RegretNet. Besides, when one item’s valuation is close to $0$ and another item's is close to $1$, the mechanism learned by RegretNet has a positive probability to allocate the item with the lower valuation to the bidder, while RegretNet-PE and the optimal mechanisms would not, as shown in Figure 3 (the printed allocation rules) in Appendix B.3.2.
>
> The results fully support our theory.

---

### Official Review · Reviewer_1aRG · 2022-07-12

**Rating:** 5
**Confidence:** 2
**Soundness:** 2 fair
**Presentation:** 2 fair
**Contribution:** 1 poor

**Summary:**

This paper considers the learning problem arising from seller revenue optimization in repeated auctions, with an approximate incentive compatibility constraint, under the assumption that the joint distribution of context and values is available to the learner (at least an empirical version). It studies whether averaging the learnt payment and allocation functions over permutations of items (resp. bidders) reduces bias error term and statistical error term when the distribution is invariant by permutation of the items (resp. bidders).

**Questions:**

How does the statistical error provided in Th. 3.6 compares to those of empirical Myerson auctions ?

**Limitations:**

Suggestions:

1. I feel the proof of Th. 3.1 could be deferred to appendix or at least shortened as it mostly comes from the definitions + convexity of max function.

2. About the case $Q_3$ (both item and user symmetry), isn't the optimal auction a second-price auction with one optimized reserve price? A sanity check could be to compare the error term of the auction learnt by the RegNet to the learning of this unique reserve price.

3. As an additional direction, the authors could compare the aggregation method to other classes of equivariant models they cite in the introduction, either theoretically or empirically (or both).

**Strengths And Weaknesses:**

1. The structure of the paper and of the exposition is clear.

2. The writing itself should be improved as a certain number of quantities are not formally defined or even not defined at all (or just in the appendix).
    * l. 125 I would have expected a more formal definition of $f(x)\sigma_m$
    * Th 3.5 introduces covering numbers ${\cal N}_{\infty,1}(\cdot,\cdot)$. I guess $\infty,1$ refers to some distance, but it is only provided in appendix
    * $v^{(l)}, x^{(l)}, y^{(l)}$ are probably samples in Th. 3.6
    * $\omega$ is again only defined in appendix (Th. 3.6)
    * $\widehat{reg}_i(\omega)$ (Th. 3.6) is not defined at all (even in appendix)
    * I guess $rgt_i = reg_i$ ?
Most of these are "_guessable_" or defined somewhere in the appendix, but remember the paper should be self-contained.

3. I find the contributions are a bit light. Indeed, Th. 3.1 follows quickly from the definition, Th. 3.6 is mostly a corollary of results in [10]. So the strongest contribution is Th. 3.5 on the dominance of covering number that comes from the projection contracting the distances. Still, in the end, the message is clear.

---

> ### Author Response · Authors · 2022-08-02
> **Response to Reviewer 1aRG (4/4)**
>
> **Q5:** *About the case $\mathcal{Q}_3$ (both item and user symmetry), isn't the optimal auction a second-price auction with one optimized reserve price? A sanity check could be to compare the error term of the auction learnt by the RegNet to the learning of this unique reserve price.*
>
> **A5:** We respectfully note that the second-price auction with one optimized reserve price is the optimal auction for **the one-item setting only**,  while this paper studies **general settings with an arbitrary number of items.** Moreover, in the one-item cases, our theory and experiments are coincident with the “second-price auction with one optimized reserve price” strategy.
>
> We further conduct additional experiments in the 2-bidder 1-item setting. Our experiment shows that equivariant neural network-based methods (RegretNet) exactly learn almost the same allocation rule with the Myerson auction, as shown in Figure 2 (printed allocation rules) in Appendix B.3.1.
>
> **Q6:** *As an additional direction, the authors could compare the aggregation method to other classes of equivariant models they cite in the introduction, either theoretically or empirically (or both).*
>
> **A6:** Thanks. We have added a detailed and comprehensive comparison in our paper.
>
> We would love to note that **all equivariant models are covered in our aggregation framework**. Every equivariant neural network can be seen as the projected network by the orbit averaging projection operator from a neural network. To clear any confusion, we will give the following proposition in Section 2, p. 4, as follows,
>
> **Proposition 2.3:** Orbit averaging $\mathcal{Q}$ is a projection to the equivariant mapping space $\{f:\psi\circ f=f\circ \rho\}$, *i.e.*, $\psi\circ\mathcal{Q}f=\mathcal{Q}f\circ\rho$ and $\mathcal{Q}^2=\mathcal{Q}$. In particular, if $f$ is already equivariant, then $\mathcal{Q}f=f$.
>
> A detailed proof is given in appendix A.1.
>
> We have also conducted comparison experiments between the EquivariantNet (a typical equivariant model obtained by modifying architecture and optimization) and Regret-PE (a equivariant model in the aggregation framework). The results fully support our theory, as shown below.
>
>
> **One bidder and two items.** The valuations are i.i.d. drawn from the uniform distribution $U[0,1]$. The optimal mechanism is proved by [19], as shown in the following table,
>
>  |     Method     | Revenue | Regret  | Optimal |
> | :------------: | :-----: | :-----: | :-----: |
> | EquivariantNet | 0.5510  | 0.00013 | 0.5500  |
> |  RegretNet-PE  | 0.5506  | 0.00040 | 0.5500  |
>
> **Two bidders and two items.** Each bidder draws the valuation for each item from the uniform distribution $U[0,1]$. We compare the optimal auction in $\text{VVCA}$ and $\text{AMA}_\text{bsym}$ families [28], as shown in the following table,
>
>  |     Method     | Revenue | Regret  | Optimal |
> | :------------: | :-----: | :-----: | :-----: |
> | EquivariantNet |  0.873  | 0.00100 |  0.860  |
> |  RegretNet-PE  |  0.874  | 0.00099 |  0.860  |
>
> From the tables, we observe that EquivariantNet and RegretNet-PE have almost the same revenues and ex-post regrets, while both of them achieve the optimal revenues with negligible ex-post regret.
>
> [9] C. Daskalakis, A. Deckelbaum, and C. Tzamos. Strong duality for a multiple-good monopolist. Econometrica, 2017.
>
> [10] Z. Duan, J. Tang, Y. Yin, Z. Feng, X. Yan, M. Zaheer, and X. Deng. A context-integrated transformer-based neural network for auction design. arXiv preprint arXiv:2201.12489, 2022.
>
> [12] P. Dutting, Z. Feng, H. Narasimhan, D. Parkes, and S. S. Ravindranath. Optimal auctions through deep learning. In International Conference on Machine Learning (ICML), 2019.
>
> [19] A. M. Manelli and D. R. Vincent. Bundling as an optimal selling mechanism for a multiple-good383
> monopolist. Journal of Economic Theory, 2006.
>
> [21] R. B. Myerson. Optimal auction design. Mathematics of operations research, 1981.
>
> [26] J. Rahme, S. Jelassi, J. Bruna, and S. M. Weinberg. A permutation-equivariant neural network architecture for auction design. In Proceedings of the AAAI Conference on Artificial Intelligence, 2021.
>
> [28] T. Sandholm and A. Likhodedov. Automated design of revenue-maximizing combinatorial auctions. Operations Research, 2015.

---

> ### Author Response · Authors · 2022-08-02
> **Response to Reviewer 1aRG (3/4)**
>
> **Q3:** *How does the statistical error provided in Th. 3.6 compares to those of empirical Myerson auctions ?*
>
> **A3:** Thanks. We guess the reviewer may refer to the error between expected revenue/regret and empirical revenue/regret (the regret is zero in empirical Myerson auctions). It is a very interesting direction! We have studied it in rebuttal from both theoretical and empirical aspects.
>
> Theoretically, we prove the following sample complexity bound for Myerson auctions:
>
> **Theorem A.3:** Assume the item valuation for each bidder is not larger than $1$. When the sample complexity satisfies $L\ge\frac{1}{2\epsilon^2}\log\frac{2}{\delta}$, with probability at least $1-\delta$, we have
>
> $
> \bigg|\frac{1}{L}\sum_{\ell=1}^Lrev(v^{(\ell)},x^{(\ell)},y^{(\ell)})-\mathbb{E}_{(v,x,y)}\Big[rev(v,x,y)\Big]\bigg|\le\epsilon.
> $
>
> We give a brief proof here. The detailed proof has been given in the appendix. Since the valuation for each bidder is not larger than $1$, the revenue is not larger than $1$. According to the Hoeffding’s inequality, we have
>
> $
> \mathbb{P}\bigg[\bigg|\frac{1}{L}\sum_{\ell=1}^L rev(v^{(\ell)},x^{(\ell)},y^{(\ell)})-\mathbb{E}_{(v,x,y)}\Big[rev(v,x,y)\Big]\bigg|\ge \epsilon\bigg]\le 2\exp(-2L\epsilon^2).
> $
>
> Let $2\exp(-2L\epsilon^2)\le\delta$, then we obtain what we need. The proof is completed.
>
> This generalization bound for the Myerson auction is smaller than the bound in Th. 3.6, which suggests that the worst case in Myerson auction has better generalizability than the worst case in equivariant neural network-based methods.
> This makes sense that learning-based method has worse worst case compared with the theoretically optimal approach. This does not necessarily imply a real worse generalizability in practice.
>
> We then conduct experiments on the 2-bidder 1-item and 3-bidder 1-item settings, where the valuations of every bidder are i.i.d. drawn from $U(0,1)$ and every experiment is repeated for 5 trials. The statistical errors of the revenue are shown below,
>
> **Two bidders and one item.**
>
> | Myerson | RegretNet-PE |
> | :-----: | :----------: |
> | 0.03138 |   0.01687    |
> | 0.01471 |   0.00546    |
> | 0.01356 |   0.00574    |
> | 0.01403 |   0.00069    |
> | 0.00474 |   0.03043    |
>
> **Three bidders and one item**
>
> | Myerson | RegretNet-PE |
> | :-----: | :----------: |
> | 0.01536 |   0.00182    |
> | 0.01835 |   0.00836    |
> | 0.01477 |   0.01704    |
> | 0.01927 |   0.01071    |
> | 0.01394 |   0.02278    |
>
> The tables show that there is no significant difference between the generalizabilities of the two approaches, which fully support our theory.
>
> **Q4:** _I feel the proof of Th. 3.1 could be deferred to the appendix or at least shortened as it mostly comes from the definitions + convexity of max function._
>
> **A4:** Thanks and addressed.

---

> ### Author Response · Authors · 2022-08-02
> **Response to Reviewer 1aRG (2/4)**
>
> **Q2:** *I find the contributions are a bit light. Indeed, Th. 3.1 follows quickly from the definition, Th. 3.6 is mostly a corollary of results in [10]. So the strongest contribution is Th. 3.5 on the dominance of covering number that comes from the projection contracting the distances. Still, in the end, the message is clear.*
>
> **A2:** We respectfully argue that our paper has made significant contributions. Please let allow us to clarify our contributions below.
>
> * **This is the first work that theoretically validates the advantages of equivariance in auction design**, which has been widely recognized as a timely and important problem with both academic and practical significance, see, _e.g._, [10, 12, 26].  We prove that
>
>   * **Permutation equivariance decreases the expected ex-post regret and increases the expected revenue** (Th. 3.1). This theorem inspires to measure the distance between a mechanism and the optimal permutation-equivariant mechanism by the difference of their ex-post regrets when the revenue is fixed. This may have values in inspiring novel algorithms for learning the target permutation-equivariant mechanism.
>
>   * **Permutation equivariance improves the generalizability** (see Th. 3.5 and Th. 3.6). This suggests that permutation-equivariant neural networks-based methods have better decision-making performance in unseen scenarios, when the training performance and the sample size are fixed. This helps design sample-efficient algorithms.
>
>   **Our theory explains previously reported experimental results** [10, 26]. Th. 3.1 explains “both CITransNet and CIEquivariantNet outperform CIRegretNet a lot in all the 3×10 and 5×10 auctions” [10]. Th. 3.5 and Th. 3.6 explain the phenomenon reported in [26] that “when fewer training samples are available, our equivariant architecture (EquivariantNet) generalizes while RegretNet struggles to.”
>
> * **We further provide an explanation for the learning process of non-permutation-equivariant neural
> networks** (NPE-NNs) (see Section 3.1.3). According to Th.3.1, every mechanism would not have a better performance than the projected permutation-equivariant mechanism via orbit averaging. Thus, intuitively, the permutation-equivariant mechanisms are the “optimal targets". In learning the optimal auction mechanism by an NPE-NN, we show that an extra positive term exists in the quadratic penalty of the ex-post regret based on the result (1). This term serves as a regularizer to penalize the “non-permutation-equivariance”. Moreover, this regularizer also interferes the revenue maximization, and thus affects the learning performance of NPE-NNs. This further explains the advantages of permutation-equivariance in auction design.

---

> ### Author Response · Authors · 2022-08-02
> **Response to Reviewer 1aRG (1/4)**
>
> Thank you for your thorough review and constructive comments. All your concerns have been carefully responded below. The manuscript is carefully revised accordingly. We sincerely hope our responses fully address your questions.
>
> **Q1:** _The writing itself should be improved as a certain number of quantities are not formally defined or even not defined at all (or just in the appendix)._
>
> **A1:** Thanks and addressed. We have carefully revised our manuscript to fix these issues. Details are given below.
>
> **Q1.1:** *l. 125 I would have expected a more formal definition of $f(x)\sigma_m$.*
>
> **A1.1:** We have added a formal definition in Section 2, p. 3, as below.
>
>   **Definition 2.2 (permutaion-equivariant mapping):** A permutation-equivariant mapping is defined to be $f:\mathbb{R}^{n\times m}\to \mathbb{R}^{n\times m}$ that for any instance $x\in\mathbb{R}^{n\times m}$, and permutation matrices $\sigma_n\in\mathbb{R}^{n\times n}$ and $\sigma_m\in\mathbb{R}^{m\times m}$, we have
>
> $f(\sigma_nx\sigma_m)=\sigma_nf(x)\sigma_m$.
>
>
> **Q1.2:** *Th 3.5 introduces covering numbers $\mathcal{N}_{\infty,1}(\cdot,\cdot)$. I guess $\infty,1$ refers to some distance, but it is only provided in the appendix.*
>
> **A1.2:** Thanks. $\infty,1$ refers to the $l_{\infty,1}$ distance, which is defined as follows,
>
> **Definition 2.4 ($l_{\infty,1}$-distance):** Let $\mathcal{X}$ be a feature space and $\mathcal{F}$ a space of some functions from $\mathcal{X}$ to $\mathbb{R}^n$. The $l_{\infty,1}$-distance on the space $\mathcal{F}$ is defined as
>
> $l_{\infty,1}(f,g)=\max_{x\in \mathcal{X}}(\sum_{i=1}^n|f_i(x)-g_i(x)|)$.
>
> This definition has been added to Section 2, p. 4, our manuscript.
>
>
>
> **Q1.3:** _$v^{(l)},x^{(l)},y^{(l)}$ are probably samples in Th. 3.6_
>
> **A1.3:** Yes, they are. To clear any confusion, we have added a formal definition in Section 2, p. 3 as below.
>
> **Definition 2.1:** Suppose the parameter of a network is $\omega$. The bidder $i$'s empirical revenue is defined as
>
> $\frac{1}{L}\sum_{l=1}^Lp^\omega_i(v^{(l)},x^{(l)},y^{(l)})$,
>
> and the ex-post regret
> $\widehat{reg}_i(\omega)$
> is defined as
>
> $\frac{1}{L}\sum_{l=1}^L{reg}_i^\omega(v^{(l)},x^{(l)},y^{(l)})$,
>
> where the set $\{(v^{(l)},x^{(l)},y^{(l)})\}_{l=1}^L$ is i.i.d. sampled from the following prior distribution as follows,
>
> $\mathbb{P}(v,x,y)=\prod_{i,j=1}^{n,m}\mathbb{P}(v_{ij}|x_i,y_j)$ $\mathbb{P}\_{X_i}(x_i)$ $\mathbb{P}\_{Y_j}(y_j)$.
>
>
> **Q1.4:** _$\omega$ is again only defined in appendix (Th. 3.6)_
>
> **A1.4:** Thanks. We have moved this definition to the main text in Section 2, p. 3.
>
>
> **Q1.5:** *$\widehat{reg}_i(\omega)$ (Th. 3.6) is not defined at all (even in appendix)*
>
>  **A1.5:** Thanks. We have added a formal definition (Definition 2.1) in Section 2, p. 3; please kindly refer to A1.3 above.
>
>
>
> **Q1.6:** *I guess $rgt_i=reg_i$ ?*
>
> **A1.6:** Thanks and addressed.

---

> ### Author Response · Authors · 2022-08-07
> **Sincerely looking forward to your reply**
>
> Dear Reviewer 1aRG,
>
> Thank you very much for your thorough comments and helpful comments. We are looking forward to your reply. If you have further concerns or requests, please kindly give us a chance to address them in the author-reviewer discussion period. If all your concerns have been resolved, it is much appreciated if you may raise the rating of our work.
>
> Thank you!
>
> The authors

---

### Meta-Review · Area_Chair_9KYr · 2022-08-24

**Recommendation:** Accept
**Confidence:** Less certain

**Metareview:**

Most reviews are in the positive direction. The reviewers gave comments on having better discussions of related work and improving presentation by making definitions more clear. I think the authors can further improve the paper based on these comments.

**Award:**

No

---

### Decision · Program_Chairs · 2022-09-14

Accept